# STING inhibits LINE-1 retrotransposition through sorting ORF1p to lysosomes for degradation

Yu Huang [1,2,5], Fengwen Xu [1,2,5], Lingwa Wang [3,5], Shan Mei [1,2], Fei Zhao [1,2], Liming Wang [1,2], Yu Xie [1,2], Liang Wei [1,2], Yamei Hu [1,2], Zhao Gao [1,2], Tiffany Xue [4], Jugao Fang [3]✉ & Fei Guo [1,2]✉

## Abstract

**The cyclic dinucleotide sensor stimulator of interferon (IFN) genes (STING) is known for its critical role in interferon and inflammatory responses. In addition, STING also has functions independent of interferon induction. In this study, we report that STING restricts the mobilization of the cellular retrotransposon long interspersed nuclear element 1 (LINE-1) independent of cGAS and interferon induction. LINE-1 is the only active autonomous retrotransposable element in the human genome and its transposition can cause genetic and autoimmune diseases. STING inhibition of LINE-1 requires its dimerization. Mechanistically, STING interacts with LINE-1 ORF1p, then the complex translocates to the ER-Golgi intermediate compartment (ERGIC) and the Golgi followed by sorting to Rab7-positive lysosomes for degradation. Our data unveil a function of STING in maintaining host genome integrity by restricting LINE-1 retrotransposition via an IFN-independent mechanism.**

Keywords STING; LINE-1; Retrotransposition; IFN-Independent; Lysosome
Subject Categories Chromatin, Transcription & Genomics; Membranes & Trafficking

## Introduction

The cGAS–STING pathway presents a major cytosolic DNA-sensing mechanism detecting pathogen DNA or abnormal cellular DNA (Li and Chen, 2018; Sun et al, 2013). The binding of cytosolic DNA by cyclic guanosine monophosphate (GMP)-adenosine monophosphate (AMP) (cGAMP) synthase (cGAS) leads to the production of second messenger cGAMP (Ma and Damania, 2016), which binds and activates the ER-resident adapter STING (also named as MITA, ERIS, and MPYS). Under steady-state conditions, STING forms a dimer. Upon cGAMP binding, the STING dimer undergoes a conformational switch and promotes the lateral

oligomerization, and then traffics to the Golgi apparatus (Burdette et al, 2011; Ishikawa and Barber, 2008). The activated STING recruits TBK1 to phosphorylate IRF3, which translocates to the nucleus to stimulate the expression of interferons and inflammatory cytokines (de Oliveira Mann et al, 2019). It is thus not surprising that gain-of-function STING mutants including V147L/M, N154S, and V155M cause severe autoinflammatory diseases called STING-associated vasculopathy with onset in infancy (SAVI) and familial chilblain lupus (G166E) (Konig et al, 2017; Patel and Jin, 2019).

STING is a 40 KDa protein, resides in the ER when not activated. It has four transmembrane (TM) helices, a ligand-binding domain in response to CDNs (cyclic dinucleotides), and a C-terminal tail (CTT) that activates TBK1 and IRF3 phosphorylation. STING is evolutionarily conserved, its homologs are found in bacteria (Morehouse et al, 2020), *Drosophila melanogaster* and *Nematostella vectensis* (Kranzusch et al, 2015). However, the CTT domain, which mediates IFN response, is only present in STING of vertebrates and mammals (Margolis et al, 2017). The ancestral STING proteins do not have CTT, but they do bind CDNs (Kranzusch et al, 2015). STING proteins of bats have the key residue S358 mutated, thus is less active in IFN induction, but can induce autophagy to defend virus infection (Xie et al, 2018). Indeed, in addition to IFN signaling, mammalian STING proteins have also been reported to modulate autophagy (Gui et al, 2019), cell death (Wu et al, 2020) and cellular homeostasis (Ranoa et al, 2019), which are independent of IFN induction.

If gain-of-function STING mutations lead to excessive IFN production and autoimmune disease SAVI, it would be expected that loss of STING subsides autoimmune diseases. Surprisingly, the absence of STING exacerbates systemic autoimmunity. For example, STING−/− MRL.Fas[lpr] mice exhibit more severe SLE disease and accelerated death. STING-knockout mice are hyperresponsive to TLR7/9 ligands, including endogenous nucleic acids (Sharma et al, 2015a), which may include reactivated retroelements that can stimulate IFN-I response and promote inflammation and autoimmunity (Mavragani et al, 2016; Sharma et al, 2015b).

Activation of retrotransposable elements can lead to insertional mutagenesis, alternative splicing, DNA damage, and genome instability (Beck et al, 2011). LINE-1 is the only autonomous

[1]Key Laboratory of Pathogen Infection Prevention and Control (Ministry of Education), State Key Laboratory of Respiratory Health and Multimorbidity, National Institute of Pathogen Biology, Chinese Academy of Medical Sciences & Peking Union Medical College, Beijing, P. R. China. [2]National Institute of Pathogen Biology and Center for AIDS Research, Chinese Academy of Medical Sciences & Peking Union Medical College, Beijing, P. R. China. [3]Department of Otorhinolaryngology, Head and Neck Surgery, Beijing Tong Ren Hospital, Capital Medical University, Beijing, China. [4]Department of Chemistry, Bryn Mawr College, Bryn Mawr, PA 19010, USA. [5]These authors contributed equally: Yu Huang, Fengwen Xu, Lingwa Wang. ✉E-mail: fangjugao2@ccmu.edu.cn; guofei@ipb.pumc.edu.cn

retrotransposon active in the human genome. It comprises about 17% of the human genome, but only 80–100 copies are mobile. The full-length LINE-1 encodes two open-reading frames (ORFs), ORF1p is a 40 KDa RNA chaperone protein, and ORF2p (150 KDa) has endonuclease and reverse transcriptase activities. Together, they associate with LINE-1 RNA to form an RNP (ribonucleoprotein) complex, transports into the nucleus, reverse transcribes LINE-1 RNA into DNA, concurrent with integration in human genome at the T/A-rich sites (5′-TTTT/AA-3′) (Cordaux and Batzer, 2009; Nicolas Gilbert et al, 2002). The human genome also has non-autonomous retrotransposons such as SINE and Alu whose mobilization depends on LINE-1 proteins (Dewannieux et al, 2003; Wallace et al, 2008).

Retrotransposition of LINE-1 and non-autonomous retrotransposons is under stringent surveillance and control by mechanisms including methylation of CpG motif in LINE-1 DNA, piRNAs and repressive histone modifications (Bodak et al, 2014; Yang and Wang, 2016). Excessive mobilization of LINE-1 disturbs cell homeostasis, causes cell senescence, cancer and autoimmune diseases (Mendez-Dorantes and Burns, 2023; Payer and Burns, 2019). This is partly because LINE-1 RNA and DNA can be ligands of nucleic acid sensors. In addition, LINE-1 ORF2p can reverse transcribe cellular RNA and synthesize abnormal cytosolic DNA, which can stimulate endosomal DNA sensor TLR9 and cytosolic DNA sensor cGAS (Bregnard et al, 2016; Mathavarajah and Dellaire, 2024; Stetson, 2012). LINE-1 RNA itself can be sensed by RIG-I/MDA5 and TLR7 (Nerbovik et al, 2017; Young et al, 2012). Activation of these nucleic acid sensors triggers IFN response, exacerbates microenvironment immune dysfunction, promotes autoimmune and inflammatory diseases (Heinrich et al, 2019). In support of this scenario, LINE-1 ORF1p and RNA levels are increased in lupus nephritis kidneys of SLE and minor salivary gland tissue of Sjogren's syndrome (SS) patients (Mavragani et al, 2016). LINE-1 ORF1p has been reported to associate with SLE autoantigens such as Ro, La and small nuclear RNP 70 (Goodier et al, 2013). There are also IgG autoantibodies against LINE-1 ORF1p in the majority of SLE patients (Carter et al, 2020). In light of the aggravation of autoimmune disease in the STING−/− mouse model and excessive activation of LINE-1 associated with SLE, a role of STING in controlling LINE-1 activity is speculated.

To investigate whether STING modulates LINE-1 activity, we examined the transcriptome of STING-knockout THP1 cells by RNA sequencing and observed an increase in expression of multiple LINE-1 elements. We further showed that STING inhibits LINE-1 through association with LINE-1 ORF1p and trafficking to lysosomes for degradation. This function of STING is independent of TBK1 and IRF3 phosphorylation and type I IFN production. These findings support a mechanism by which STING subdues autoimmune diseases through suppressing the activity of LINE-1 and other transposable elements.

# Results

## STING inhibits LINE-1 retrotransposition

To examine the effect of STING on cellular gene expression, we knocked out STING in THP1 cells and performed RNAseq to characterize the transcriptome. About 2233 differentially expressed genes (DEGs) were identified, with 824 upregulated and 1409 downregulated (Fig. EV1A). Gene ontology (GO) analysis of the downregulated genes revealed that among the top 20 enriched biological processes, some are known to involve STING, including inflammation, carbohydrate derivative catabolic process and second messenger-mediated signaling. The upregulated genes indicate enriched biological processes predominantly linked with neuron projection development, epithelial migration and proliferation and leukocyte migration (Fig. EV1B). Specially, positive regulation of defense response was a highly enriched, upregulated clustering in STING knockout cells. Results of KEGG analysis showed that STING knockout markedly impacted pathways involved in the immune system, endocrine system, nervous system, cancers, infectious disease, cardiovascular disease, endocrine and metabolic disease and neurodegenerative disease (Fig. EV1C). Opposite effects of STING knockout on the expression of IFN-stimulated genes (ISGs) and Inflammation-related factors were observed, with some downregulated, such as ZNF, IFITM1, IFI6, CCL22, and CXCL16, and some upregulated, such as IL6R, IL3R, CD86, STAT4, and IFI16 (Fig. EV1D). We further sequenced small RNA (sRNA) with the length of 18 to 30 nt in STING knockout THP1 cells. A twofold increase was observed for sRNA molecules that are mapped to DNA transposons and RNA retroelements (Fig. EV1E).

Further interrogation of RNAseq data showed an increase in the expression of 17 out of 26 classes full-length LINE-1 RNA in STING-knockout THP1 cells (Fig. 1A). We next performed RT-qPCR to determine levels of LINE-1 RNA using primers amplifying three regions in LINE-1, the 5′ UTR, ORF1, and ORF2, and the results confirmed the significant increase in levels of endogenous LINE-1 RNA in STING knockout THP1 cells (Fig. 1B,C). These results suggest that STING suppresses LINE-1 activity.

To quantify the effect of STING on LINE-1 activity, we utilized an established CMV-L1-neo$^{RT}$ reporter system which contains a neomycin resistance gene in 3′ UTR region of LINE-1 (Esnault et al, 2000; Moran et al, 1996). The neomycin resistance gene only expresses after successful retrotransposition of L1-neo$^{RT}$ (Fig. 1D). We co-transfected CMV-L1-neo$^{RT}$ reporter DNA and STING-Flag DNA into HeLa cells, and monitored LINE-1 activity by scoring G418-resistant cell colonies. STING exerted dose-dependent inhibition of LINE-1 retrotransposition (Fig. 1E), but did not affect the number of G418-resistant colonies produced by pEGFP-N1 vector which expresses the neomycin resistance gene (Fig. 1F). Overexpression of STING did not affect cell viability (Fig. 1G). Next, we knocked down STING with siRNA in HeLa cells (Fig. 1H), and observed significant increase of CMV-L1-neo$^{RT}$ reporter activity (Fig. 1I), no effect on pEGFP-N1-generated G418-resistant colonies (Fig. 1J). The pEGFP-N1 colony results suggest that STING overexpression or knockdown does not affect CMV-promoter-mediated transcription of the neomycin resistance gene. Then, we generated HeLa cell lines that either stably express STING or have STING knocked out with CRISPR/Cas9. Consistent with the CMV-L1-Neo$^{RT}$ data of transient expression or knockdown of STING, a lower number of CMV-L1-Neo$^{RT}$ colonies was scored in the STING-expressing cell line, and a higher number in the STING-knockout cell line (Fig. 1K,L). Together, these data demonstrate that STING inhibits LINE-1 retrotransposition.

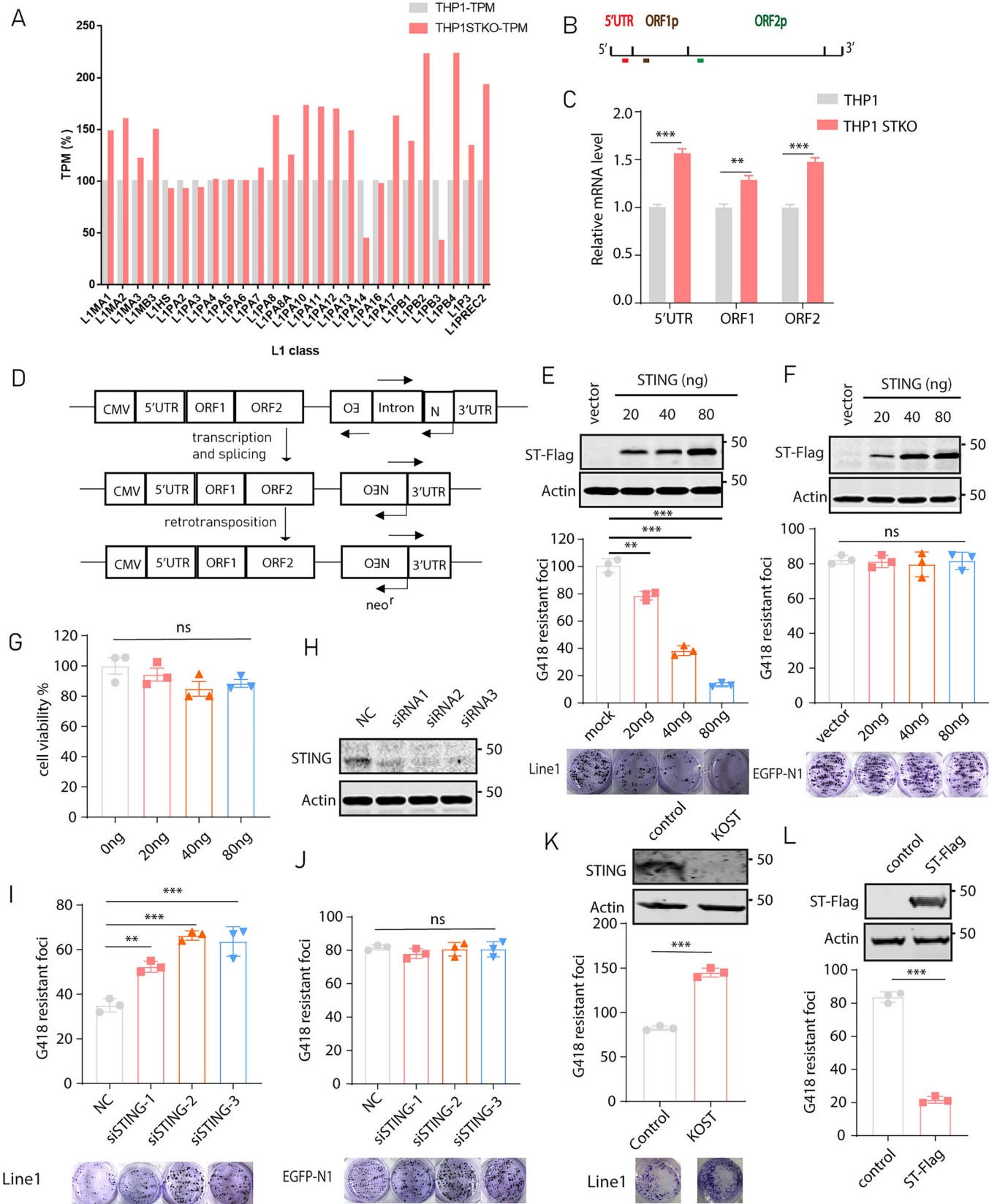

**Figure 1. STING inhibits LINE-1 retrotransposition.**

(A) Relative transcripts per kilobase of exon model per million mapped reads (TPM) of LINE-1 elements whose lengths exceed 6Kb in the GRCh38.p14 reference genome in THP1 STING knockout cell line. (B) The location of the amplified regions in the LINE-1 genome was shown. Red, 5′UTR; brown, ORF1; green, ORF2. (C) Endogenous LINE-1 mRNA in the THP1 STING-knockout cell line was examined with RT-qPCR. The results of three independent experiments are presented in the bar graph. Compared with the THP1 control cell line, significant differences were observed with the STING knockout cell line in 5′UTR ($p < 0.001$), ORF1 ($p = 0.003$), and ORF2 ($p < 0.001$). (D) Illustration of the CMV-L1-neo$^{RT}$ reporter plasmid. Following transcription from the 5′ UTR promoter of LINE-1, the intron in the neomycin resistance gene is excised. This intronless mRNA is then reverse transcribed into cDNA that is able to produce a functional neomycin resistance mRNA. (E) HeLa cells were transfected with CMV-L1-neo$^{RT}$ together with different doses of STING-Flag DNA. G418-resistant cell colonies were scored, and the results of three independent experiments are presented in the bar graph. Compared with the mock vector control, significant differences were observed at 20 ng ($p = 0.002$), 40 ng ($p < 0.001$), and 80 ng ($p < 0.001$) in the STING-Flag transfection group. Images of representative colony assays are shown. Ectopic expression of STING-Flag was examined by Western blots. (F) HeLa cells were transfected with the pEGFP-N1 plasmid together with different doses of STING-Flag DNA. G418-resistant cell colonies were scored and results of three independent experiments are presented in the bar graph. Compared with the mock vector control, no significant difference was observed at 20 ng ($p = 0.988$), 40 ng ($p = 0.839$), and 80 ng ($p = 0.996$) in the STING-Flag transfection group. Images of representative colony assays are shown. Ectopic expression of STING-Flag was examined by Western blots. (G) Cell viability of HeLa cells with different doses of STING-Flag expression. Compared with the mock vector control, no significant difference was observed at 20 ng ($p = 0.695$), 40 ng ($p = 0.099$), and 80 ng ($p = 0.227$) in the STING-Flag transfection group. (H) siRNA was used to knockdown endogenous STING in HeLa cells. STING expression was examined with Western blots. (I) Colony assay was performed to determine CMV-L1-neo$^{RT}$ reporter retrotransposition activity with STING knockdown by siRNA in HeLa cells. Compared with the negative control (NC), significant differences were observed in siRNA1 ($p = 0.002$), siRNA2 ($p < 0.001$), and siRNA3 ($p < 0.001$). (J) HeLa cells were transfected with the pEGFP-N1 plasmid DNA after STING knockdown by siRNA. G418-resistant colonies were scored and results of three independent experiments are presented in the bar graph. Compared with the negative control (NC), no significant difference was observed in siRNA1 ($p = 0.474$), siRNA2 ($p = 0.970$), and siRNA3 ($p = 0.970$). (K) STING knockout HeLa cell line was transfected with 250 ng CMV-L1-neo$^{RT}$ reporter plasmid. Forty-eight hours post transfection, G418 was added to the medium and maintained for 12 days. G418-resistant colonies were scored, and the results of three independent experiments are presented in the bar graph (mean ± SEM; paired $t$-test). Compared with the control cell line, a significant difference was observed in the STING-knockout cell line ($p < 0.001$). The protein level of endogenous STING was determined with Western blots, the result was shown above the bar graph. (L) STING stably overexpressing HeLa cell line was transfected with 250 ng CMV-L1-neo$^{RT}$ reporter DNA. G418-resistant cell colonies were scored, and the results of three independent experiments are presented in the bar graph (mean ± SEM; paired $t$-test). Compared with the control cell line, a significant difference was observed in the STING-Flag stably-expressing cell line ($p < 0.001$). (E–G, I–L) Data were the average from three independent experiments (mean ± SEM; paired $t$-test). ns non-significant; $*p < 0.05$; $**p < 0.01$; $***p < 0.001$. Source data are available online for this figure.

## STING decreases LINE-1 ORF1p expression and the number of γH2AX induced by LINE-1 retrotransposition

Next, we determined which step of LINE-1 retrotransposition is affected by STING. First, we co-transfected CMV-L1-neo$^{RT}$ and STING-Flag DNA into HeLa cells, and observed a decrease in ORF1 protein (Fig. 2A) as well as in reverse transcribed LINE-1 DNA (Appendix Fig. S1). When endogenous STING was knocked down by siRNA, level of ORF1 protein from transiently transfected CMV-L1-neo$^{RT}$ increased (Fig. 2B). We next measured levels of endogenous LINE-1 mRNA of 5′UTR, ORF1 and ORF2 by qRT-PCR when STING was transiently overexpressed or knocked down by siRNA, did not observe significant difference compared to control vector (Fig. 2C,D). Knockdown of STING in THP1 cells similarly resulted in an upregulation of endogenous ORF1p (Appendix Fig. S2A), while its mRNA expression remained unchanged (Appendix Fig. S2B). However, in the HeLa cell line stably-expressing STING-Flag, endogenous ORF1 protein (Fig. 2E) and LINE-1 mRNA were decreased compared with the control cell line (Fig. 2F), which is corroborated by the increases in the STING-knockout cell line (Fig. 2G,H). We also introduced a 5′UTR promoter reporter system based on firefly luciferase activity to further investigate whether STING affects the transcription of LINE-1 (Fig. 2I). The 5′UTR promoter reporter DNA was transfected with STING-Flag plasmid into HeLa cells, and the luciferase activity was measured 48 h post transfection. There was also no significant difference compared to the control vector for LINE-1 5′UTR promoter activity (Fig. 2I). These results suggest that STING inhibits LINE-1 activity and reduces ORF1 protein expression but not transcription. The changes in endogenous LINE-1 RNA levels between the STING-expressing or STING-knockout cell lines and control cells (Fig. 2F,H) might be a result of the overall decline in the retrotransposition of endogenous LINE-1.

Since LINE-1 ORF2 protein has endonuclease activity and is able to cut cellular DNA, which leads to accumulation of γH2AX

foci as part of the DNA damage repair response, we reasoned that STING suppression of LINE-1 activity should diminish the number of γH2AX foci in the nucleus. Indeed, in the STING-knockout cell line that was transfected with CMV-L1-neo$^{RT}$ DNA, a greater number of γH2AX foci in the nucleus was detected compared to that in the vector plasmid-transfected control (Fig. 2J–L), in agreement with the decrease in γH2AX foci number in STING-overexpressing cells (Fig. EV2A–C). These data suggest that STING suppresses the formation of double-stranded DNA breaks as a result of inhibiting LINE-1 retrotransposition, thus protects host genome.

## STING inhibition of LINE-1 is independent of cGAS and IFN production

L1 elements are transcribed into RNA that is reverse transcribed into DNA, which is inserted into the genome at a new site. Activated L1 increases genome damage and cytoplasmic DNA levels, which can activate cGAS–STING and induce type I interferon (IFN) signaling. Furthermore, cGAS (Zhen et al, 2023) and some IFN-stimulated gene (ISG) products have been reported to regulate LINE-1 retrotransposition. We therefore investigated whether cGAS is required for STING to inhibit LINE-1. The cGAS-knockout HeLa cell line was used to test whether sensing of LINE-1 by cGAS is important for STING inhibition of LINE-1. We transfected STING-Flag and CMV-L1-neo$^{RT}$ plasmid to cGAS-knockout or control HeLa cells (V2) and scored the number of cell colony which reports LINE-1 activity. The results showed that STING remains inhibiting LINE-1 retrotransposition when cGAS was knocked out (Fig. 3A). ORF1 protein level was decreased in cGAS-knockout cells when STING was overexpressed (Fig. 3B). These results suggest that LINE-1 inhibition by STING is independent of cGAS.

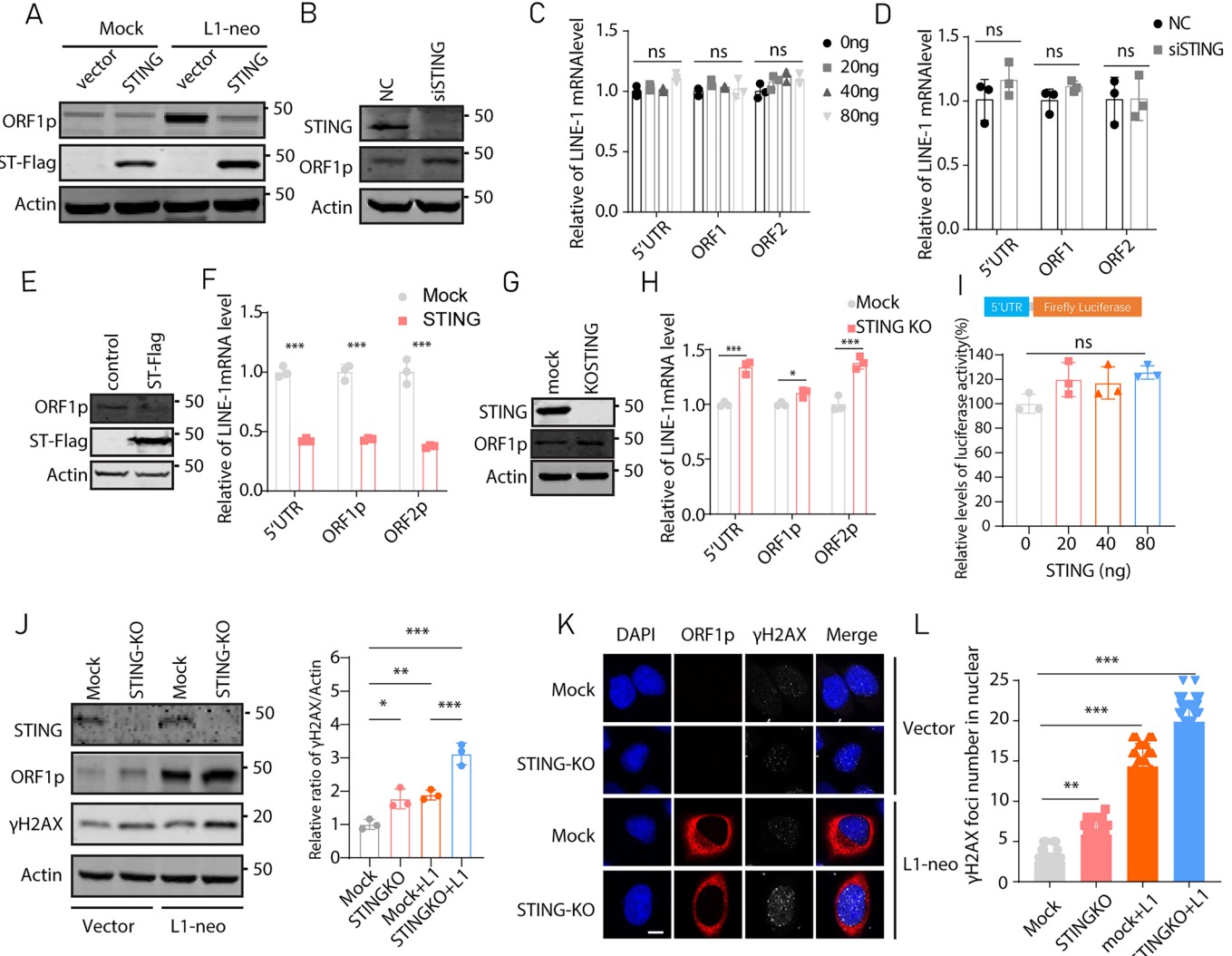

Since STING is a master regulator of IFN response and some of the IFN-stimulated gene (ISG) products such as MxB (Goodier et al, 2015; Huang et al, 2022), APOBEC3 (Arias et al, 2012; Schumann, 2007), TREX-1 (Garcia Perez and Alarcon-Riquelme, 2017; Stetson et al, 2008) and ZAP (Goodier et al, 2015; Moldovan and Moran, 2015) have been reported to curtail LINE-1 retrotransposition, it is possible that STING suppression of LINE-1 is a result of activation of these ISGs. To test this, we generated the STING 1–340 mutant, which lacks the STING C-terminal tail (CTT) and consequently loses the ability of phosphorylating IRF3 and inducing IFN production (Yin et al, 2012) (Fig. 3C,D). We co-transfected CMV-L1-neo^RT with STING wildtype (STING WT) or the 1–340 mutant into HeLa cells and scored cell colonies reporting LINE-1 activity. The result showed that STING 1–340 inhibited LINE-1 retrotransposition and ORF1 protein expression as efficiently as wild-type STING (Fig. 3E,F). Two more STING mutants S358A and S366A were tested in LINE-1 assays, which are defective in phosphorylating IRF3 (Liu et al, 2015; Tanaka and Chen, 2012; Zhao et al, 2016; Zhong et al, 2008) (Fig. 3G), and both were capable of inhibiting LINE-1 activity and ORF1 protein

expression from the CMV-L1-neo^RT reporter (Fig. 3H,I). Furthermore, we knocked down endogenous TBK1 with siRNA to block IRF3 phosphorylation and IFN production in HeLa cells (Fig. 3J), and STING still efficiently inhibited LINE-1 activity (Fig. 3K). These results demonstrate that STING inhibition of LINE-1 is independent of IFN response.

## STING inhibition of LINE-1 retrotransposition requires STING exit from the ER

Next, we investigated whether the known functional motifs and properties of STING are required for inhibiting LINE-1. STING has the N-terminal four transmembrane (TM) helices that allow localization at the ER, a cytosolic ligand-binding domain (LBD), and a C-terminal tail (CTT) that initiates IFN response (Fig. 4A). We have shown that deletion of the CTT domain does not affect STING inhibition of LINE-1. ER-bound STING forms a dimer under the steady-state condition (Ouyang et al, 2012; Shang et al, 2019). We therefore tested the effects of STING mutant △153–173 (defective in dimerization) (Ouyang et al, 2012), RIR/AIA

**Figure 2.  STING reduces LINE-1 ORF1p expression and the number of γH2AX foci induced by LINE-1 retrotransposition.**

(A) Levels of ORF1 protein were determined by Western blots in HeLa cells, which were co-transfected with CMV-L1-neo[RT] DNA and STING-Flag for 24 h. (B) siRNA was used to knockdown endogenous STING in HeLa cells. Endogenous ORF1p was detected by Western blot. (C) Endogenous mRNA levels of LINE-1 5′UTR, ORF1 and ORF2 in HeLa cells transfected with STING-Flag were measured by RT-qPCR. There was no significant difference at 20 ng ($p = 0.997$), 40 ng ($p = 0.999$), and 80 ng ($p = 0.056$) compared with 0 ng STING-Flag transfection in the 5′UTR group. There was no significant difference at 20 ng ($p = 0.611$), 40 ng ($p = 0.977$), and 80 ng ($p = 0.999$) compared with 0 ng STING-Flag transfection in the ORF1 group. There was no significant difference at 20 ng ($p = 0.335$), 40 ng ($p = 0.057$), and 80 ng ($p = 0.133$) compared with 0 ng STING-Flag transfection in the ORF2 group. (D) Endogenous mRNA levels of LINE-1 5′UTR, ORF1, and ORF2 in HeLa cells transfected with siRNA targeting STING were measured by RT-qPCR. Compared with NC, a significant difference was observed in the STING siRNA knockdown group of 5′UTR ($p = 0.479$), ORF1 ($p = 0.717$), and ORF2 ($p = 0.999$). (E) Endogenous LINE-1 ORF1p were detected with Western blots in the STING knockout HeLa cell line. (F) Endogenous LINE-1 mRNA was determined by RT-qPCR with primers of 5′UTRF/5′UTRR, ORF1pF/ORF1pR, and ORF2pF/ ORF2pR in HeLa cells stably-expressing STING-Flag. Compared with the mock cell line, a significant difference was observed in the STING-Flag stably-expressing cell line of 5′UTR ($p < 0.001$), ORF1 ($p < 0.001$), and ORF2 ($p < 0.001$). (G) Endogenous LINE-1 ORF1p were detected with Western blots in STING knockout HeLa cells. (H) Endogenous LINE-1 mRNA was determined by RT-qPCR in the STING knockout HeLa cell line. Compared with the mock cell line, a significant difference was observed in the STING-knockout cell line of 5′UTR ($p < 0.001$), ORF1 ($p = 0.048$), and ORF2 ($p < 0.001$). (I) Schematic of pGL3-5′-UTR-Luciferase (5UTR-Luc) reporter plasmid. HeLa cells were co-transfected with STING-Flag and 5′UTR-Luc reporter plasmids. Firefly luciferase activity was measured to represent the activity of the LINE-1 5′-UTR promoter 48 h post transfection. The TK-RLuc plasmid DNA was included in each transfection to control the efficiency of different transfections. The results shown are the ratios of firefly luciferase activity versus *Renilla* luciferase activity. Compared with the mock vector control, no significant difference was observed at 20 ng ($p = 0.125$), 40 ng ($p = 0.197$), and 80 ng ($p = 0.056$) in the STING-Flag transfection group. (J) Western blots to detect γH2AX in STING knockout HeLa cells transfected with CMV-L1-neo[RT]. Protein expression levels were quantified by scanning the Western blot bands and determining the grayscale intensity using ImageJ. Compared with the mock cell line, a significant difference was observed in the STINGKO cell line ($p = 0.021$) and the mock cell line with L1 transfection ($p = 0.009$). A significant difference was observed in the STINGKO cell line with L1 transfection ($p < 0.001$) compared with mock L1 transfection. (K, L) Detection of γH2AX foci in the control and STING knockout HeLa cells transfected with CMV-L1-neo[RT]. Immunofluorescence was performed 24 h post transfection (J). Scale bar, 10 µm. γH2AX foci were scored in 50 cells, the results are presented in (K) (mean ± SEM; paired *t*-test). Compared with the mock cell line, a significant difference was observed in the STINGKO cell line ($p = 0.003$), the mock cell line with L1 transfection ($p < 0.001$) and the STINGKO cell line with L1 transfection ($p < 0.001$). (C, D, F, H–J, L) Data were the average from three independent experiments (mean ± SEM; paired *t*-test). ns non-significant; *$p < 0.05$; **$p < 0.01$; ***$p < 0.001$. Source data are available online for this figure.

(defective in ER location) (Sun et al, 2009) and RY238/240AA (defective in CDN binding) (Gui et al, 2019; Zhang et al, 2019) on its ability to suppress LINE-1 retrotransposition activity. All three mutants lost the ability of inducing IRF3 phosphorylation (Fig. 4B). In the LINE-1 colony assay, as opposed to STING mutants S358A and S366A which are defective in inducing TBK1 and IRF3 phosphorylation but inhibited LINE-1 to the same degree as wild-type STING, STING mutants △153–173, RIR/AIA and RY238/ 240AA were all attenuated in LINE-1 inhibition (Fig. 4C), which was corroborated by their neutral effects on ORF1 protein expression (Fig. 4C). In agreement with these observations, wild-type STING and its mutants S358A and S366A, but not △153–173, RIR/AIA and RY238/240AA, reduced γH2AX foci which were induced by the L1-neo[RT] reporter (Fig. 4D). These data suggest that ER localization, dimerization and cGAMP binding site are required for STING-mediated LINE-1 suppression.

Since STING constitutively localizes to the endoplasmic reticulum (ER) as a dimer under physiological conditions, the requirement of its ER-targeting RIR motif and dimerization motif (residues 153–173) for LINE-1 suppression is mechanistically plausible. However, given that STING-mediated LINE-1 inhibition occurs independently of cGAS, the observed impact of the cGAMP-binding site on this activity presents an intriguing paradox. We therefore sought to investigate whether STING-dependent LINE-1 suppression requires the presence of cGAMP. To avoid the impact of cGAMP on STING activity, we generated a HeLa cell line that stably expresses STING RY238/240AA and assessed the expression of endogenous ORF1p. The results showed that the endogenous ORF1p expression level in the STING RY238/240AA cell line was comparable to that in the control cells (Fig. EV3A), indicating that the STING mutant RY238/240AA does not downregulate endogenous ORF1p. Additionally, LINE-1 activity from the transfected CMV-L1-neo[RT] reporter system was not affected in the STING RY238/240AA stable cell line (Fig. EV3B). To further exclude

cGAMP production, cGAS-knockout cells were utilized to over-express STING along with the CMV-L1-neo[RT] reporter system. Consistent with previous findings, STING significantly suppressed ORF1p expression and LINE-1 retrotransposition, while the RY238/240AA mutant lost this inhibition (Fig. EV3C,D). These results suggest that LINE-1 inhibition of STING requires the RY238/240 motif, not the presence of cGAMP.

Since STING trafficking via ERGIC to Golgi is central for signal transduction (Dobbs et al, 2015; Gonugunta et al, 2017; Gui et al, 2019; Saitoh et al, 2009), we examined STING trafficking in the context of LINE-1. In HeLa cells, stably expressed STING was mainly detected in the ER (Fig. 4E). In cells transfected with CMV-L1-neo[RT] DNA, STING left the ER, and colocalized with ERGIC, and further trafficked to the Golgi (Fig. 4E). Coincidentally, ORF1p colocalized with STING in these membrane compartments (Fig. 4E), and colocalization was presented with a fluorescence intensity curve. When we used brefeldin A (BFA) to disrupt the Golgi, STING and ORF1p were sequestered at ER and ERGIC but not colocalized with Golgi (Fig. 4E), and LINE-1 ORF1 protein expression was less inhibited by STING under BFA treatment (Fig. 4F). These results suggested that STING-mediated degradation of ORF1p requires a trafficking process from the ER to the Golgi.

Furthermore, STING colocalized with ER, ERGIC and Golgi markers in cGAS-knockout cells that were co-transfected with STING-EGFP and CMV-L1-neo[RT], as shown by immunofluorescence analysis (Appendix Fig. S3). These findings further support that the trafficking of STING, upon its interaction with LINE-1 ORF1p, is independent of cGAS and cGAMP.

## STING-mediated degradation of LINE-1 ORF1p is independent of the autolysosome

Co-trafficking of STING and ORF1p prompted us to further examine whether these two proteins interact. Indeed, the stably expressed STING-Flag was co-immunoprecipitated with

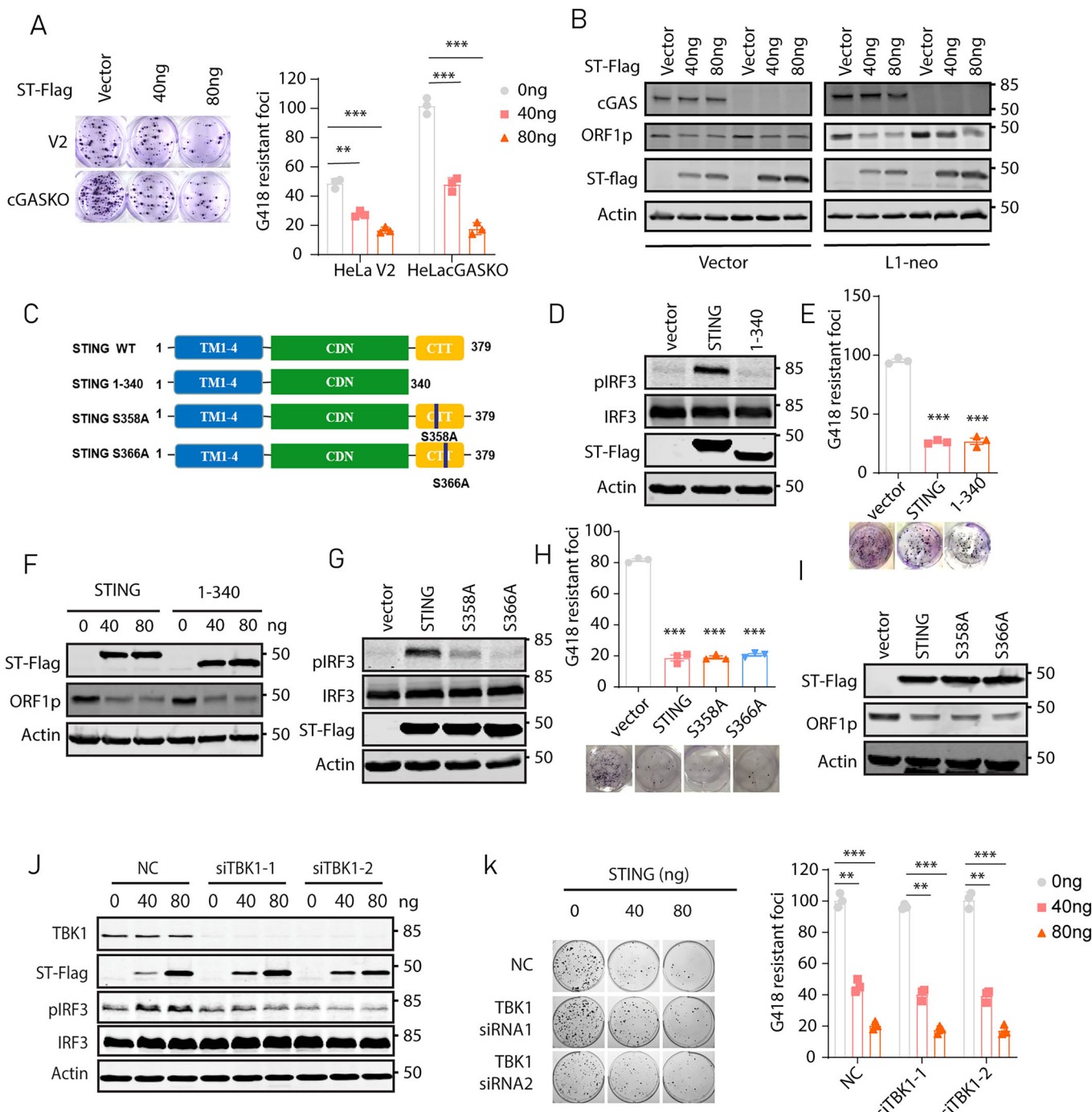

endogenous ORF1p (Fig. 5A). Moreover, STING-Flag was co-immunoprecipitated with ORF1p that was expressed from CMV-L1-neo^RT (Fig. 5A). This interaction was RNA- and cGAMP-independent (Fig. 5B,C), but required STING dimerization domain 153–173 and RY238/240 cGAMP-binding site (Fig. 5D,E). We also observed that stably expressed STING-Flag colocalized with ORF1p (Fig. 5F).

We next examined how STING decreases LINE-1 ORF1p expression, via proteasomes or lysosomes. We co-transfected STING-Flag with CMV-L1-neo^RT into HeLa cells, treated with proteasome or lysosome inhibitors, and measured LINE-1 ORF1p levels with immunoblotting. We found that ORF1p reduction was effectively blocked when cells were treated with lysosome inhibitors BafA1 or chloroquine (CQ) (Fig. 5G), suggesting that lysosomes play an important role in STING-mediated ORF1p degradation. In contrast, proteasome inhibitor MG132 did not rescue STING-mediated decrease of ORF1p (Fig. 5H). These data suggest that STING-mediated ORF1p degradation is dependent on lysosomes.

STING activation has been shown to induce autophagy, leads to formation of STING and LC3 puncta, and this process is dependent

**Figure 3. STING inhibition of LINE-1 retrotransposition is independent of cGAS and IFN response.**

(A) HeLa cGAS knockout (cGASKO) and control cell lines (V2) were transfected with CMV-L1-neo[RT] reporter plasmid and different doses of STING-Flag. G418-resistant cell colonies were scored. The results of three independent experiments are presented in the bar graph (mean ± SEM; paired $t$-test). In the mock cell line, significant differences were observed at 40 ng ($p = 0.004$) and 80 ng ($p < 0.001$) STING-Flag overexpression compared with 0 ng. In the cGAS-knockout cell line, significant differences were observed at 40 ng ($p < 0.001$) and 80 ng ($p < 0.001$) STING-Flag transfection compared with 0 ng. $*p < 0.05$; $**p < 0.01$; $***p < 0.001$. (B) ORF1 protein expression was detected by Western blots in cGAS knockout or control HeLa cells co-transfected with CMV-L1-neo[RT] and STING-Flag. (C) Illustration of wildtype (WT) and CTT-deleted STING. (D) Detection of phosphorylated IRF3 in HeLa cells transfected with STING and STING 1-340 mutant at 24 h. (E) CMV-L1-neo[RT] colony assay to measure the effect of wild-type STING or its 1–340 mutant on LINE-1 retrotransposition in HeLa cells. The results of three independent experiments are presented in the bar graph (mean ± SEM; paired $t$-test). Compared with the vector control, a significant difference was observed in STING ($p < 0.001$) and 1–340 mutant ($p < 0.001$). $*P < 0.05$; $**P < 0.01$; $***P < 0.001$. (F) ORF1p protein was detected by Western blots in HeLa cells co-transfected with CMV-L1-neo[RT] and wild-type STING or its 1–340 mutant. (G) Phosphorylation of IRF3 in response to wild-type STING, or its S358A and S366A mutants in HeLa cells. (H) CMV-L1-neo[RT] colony assay to measure the effect of wild-type STING or its S358A and S366A on LINE-1 retrotransposition in HeLa cells. G418-resistant cell colonies were scored. The results of three independent experiments are presented in the bar graph (mean ± SEM; paired $t$-test). Compared with the vector control, a significant difference was observed in STING ($p < 0.001$), S358A ($p < 0.001$), and S366A mutant ($p < 0.001$). $*P < 0.05$; $**P < 0.01$; $***P < 0.001$. (I) ORF1p protein was detected by Western blots in HeLa cells co-transfected with CMV-L1-neo[RT] and wild-type STING or its S358A and S366A mutants. (J) Phosphorylation of IRF3 was detected in HeLa cells when TBK1 was knocked down by siRNA and STING-Flag was overexpressed. (K) CMV-L1-neo[RT] colony assay was performed in TBK1 knockdown HeLa cells, which were co-transfected with CMV-L1-neo[RT] DNA and STING-Flag plasmid. G418-resistant cell colonies were scored. The results of three independent experiments are presented in the bar graph (mean ± SEM; paired $t$-test). Compared with the 0 ng STING-Flag transfection, significant differences were observed at 40 ng ($p = 0.004$) and 80 ng in NC group; in siTBK1-1 group, compared with 0 ng STING-Flag transfection, significant differences were observed at 40 ng ($p = 0.003$) and 80 ng ($p < 0.001$); in siTBK1-2 group, compared with the 0 ng STING-Flag transfection, significant differences were observed at 40 ng ($p = 0.004$) and 80 ng ($p < 0.001$). ns non-significant; $*p < 0.05$; $**p < 0.01$; $***p < 0.001$. Source data are available online for this figure.

on ATG5 and WIPI2 but not phosphorylation of TBK1 and IRF3 (Gui et al, 2019; Wan et al, 2023). To test whether STING inhibition of LINE-1 depends on induction of autophagy, we mutated amino acids L333/R334 (LR333/334AA), which are essential for STING to induce autophagy (Gui et al, 2019), and tested this mutant in the LINE-1 colony assay. The results showed that the autophagy-defective mutant LR333/334AA still strongly inhibited LINE-1 retrotransposition and ORF1p expression (Fig. 5I,J). Next, we performed an immunofluorescence experiment to examine whether STING-induced LC3 puncta contain LINE-1 ORF1p, and did not observe colocalization of STING and ORF1p in LC3B-positive autophagosomes (GFP-labeled or endogenous LC3) with or without BafA1 treatment (Fig. 5K; Appendix Fig. S4). We further knocked out ATG5 in HeLa cells to block STING-induced autophagy and measured the effect on CMV-L1-neo[RT] retrotransposition activity. Results of the LINE-1 reporter assay showed that STING inhibition of LINE-1 was not affected by ATG5 knockout (Fig. 5L). ATG5 knockout did not restore ORF1p expression (Fig. 5M). The same result was obtained in ATG5 knockdown HeLa cells (Fig. EV4A). Furthermore, we used siRNA to knockdown other STING-dependent autophagy factors ULK1 (Gui et al, 2019) (Fig. EV4B), ATG9a (Saitoh et al, 2009) (Fig. EV4C), P62 (Prabakaran et al, 2018) (Fig. EV4D), or Beclin1 (Liu et al, 2019) (Fig. EV4E). None of these knockdowns rescued STING-mediated inhibition of LINE-1 retrotransposition and decrease of ORF1p expression. We observed that knocking down ATG5 or ULK1 increased CMV-L1-neo[RT] retrotransposition, this suggests the significant role of macroautophagy in LINE-1 regulation. Together, these data suggest that autophagy is dispensable for STING-mediated LINE-1 inhibition.

## STING-mediated LINE-1 ORF1p degradation requires sorting to endolysosomes

We further investigated the role of lysosomes in STING-mediated degradation of LINE-1 ORF1p. Two patches of negatively charged residues, E282/D283 and E296/D297, have been implicated in STING trafficking to lysosomes. We first constructed E282A/

D283A and E296A/D297A mutants, and found that E282A/D283A lost the ability of inhibiting LINE-1 retrotransposition (Fig. 6A) and decreasing ORF1p expression (Fig. 6B). STING E282A/D283A mutant did not diminish the number of γH2AX foci in nucleus induced by LINE-1 retrotransposition (Fig. 6C). To confirm that STING-mediated degradation of ORF1p takes place in lysosomes, we tagged ORF1p with both mCherry (pH insensitive sensor) and EGFP (pH-sensitive sensor), and transfected the plasmid into HeLa cells stably-expressing STING-Flag. In the control cells, ORF1p was detected as yellow fluorescent signals. With expression of STING-Flag, ORF1p was detected mainly as red fluorescent signals with very weak green fluorescence, suggesting localization of ORF1p in lysosomes for degradation (Appendix Fig. S5).

Recent studies reported that STING is degraded by the endosomal sorting complexes required for transport (ESCRT)-driven lysosomal microautophagy to prevent hyperactivation (Balka et al, 2023a; Gentili et al, 2023; Kuchitsu et al, 2023). We silenced the ESCRT nucleating factors TSG101 and ALIX to explore the role of STING in LINE-1 regulation (Appendix Fig. S6A). STING retained its capacity to reduce ORF1p levels (Appendix Fig. S6B,C) and inhibit CMV-L1-neo[RT] reporter retrotransposition (Appendix Fig. S6D,E) upon either single or combined knockdown of TSG101 and ALIX. These results indicate that LINE-1 ORF1p degradation by STING is independent of ESCRT-related lysosomal microautophagy.

Lastly, we performed immunofluorescence staining to detect colocalization of STING and ORF1p by lysosomal markers Rab7 and LAMP1. We observed colocalization of ORF1p/STING with the late endosome marker Rab7 and the lysosome marker LAMP1, which was enhanced by BafA1 treatment, suggesting that the ORF1p/STING complex is sorted to lysosomes for degradation (Fig. 6D,E). In agreement with the previous report (Gonugunta et al, 2017), the STING E282A/D283A mutant induced IRF3 phosphorylation and LC3 lipidation (LC3II) (Fig. EV5A), interacted with ORF1p (Fig. EV5B) but failed to co-localize with Rab7 and LAMP1 with or without BafA1 treatment (Fig. EV5C,D). Together, these data demonstrate that STING suppresses LINE-1 activity by sorting ORF1p to lysosomes for degradation, and amino acids E282/D283 are central for this function of STING.

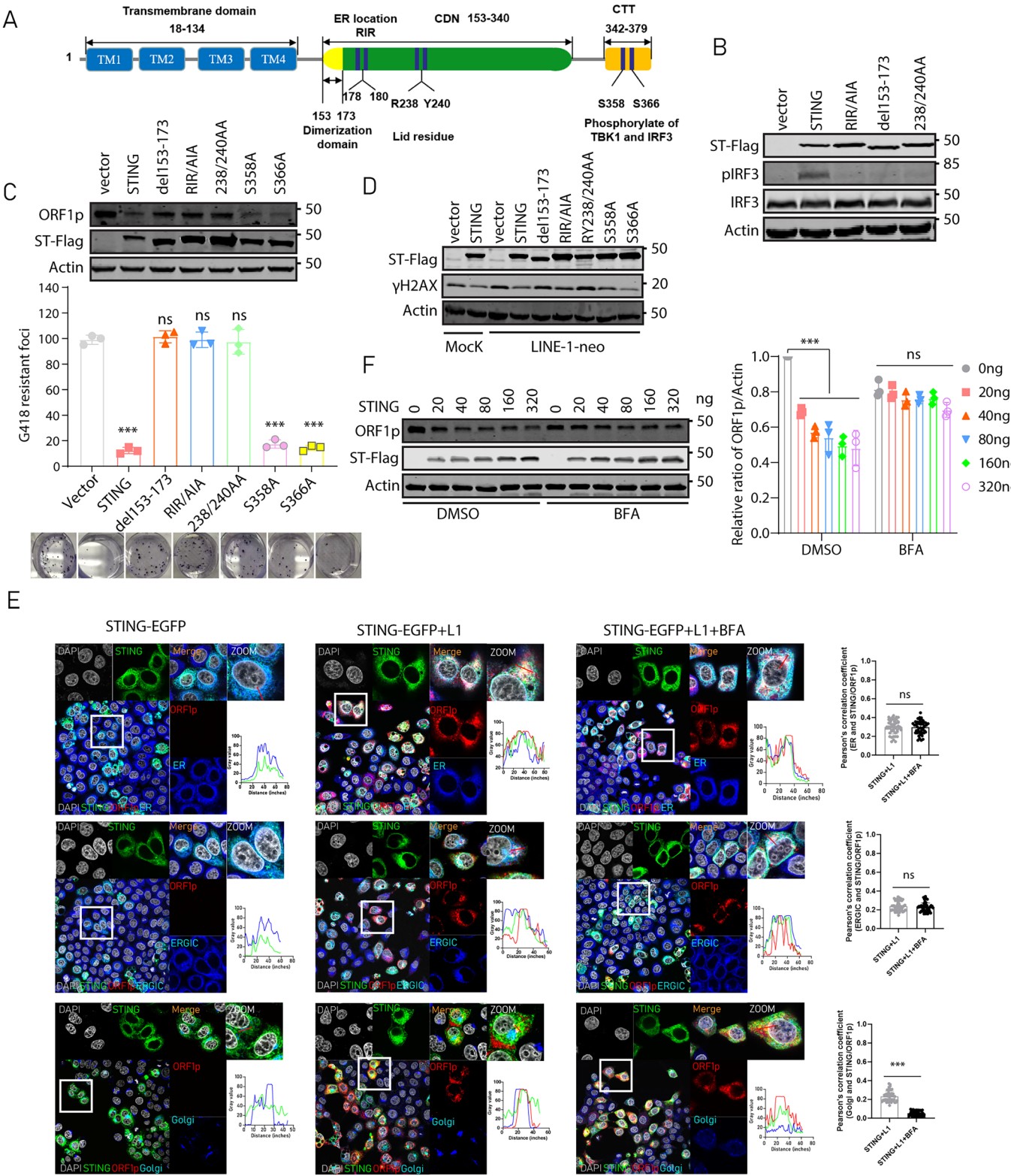

## Discussion

In this study, we demonstrate that STING suppresses LINE-1 activity by targeting ORF1p to lysosomes for degradation, thus unveiling a new layer of the multifaceted cellular mechanisms controlling LINE-1, which is the only active autonomous retroelement in the human genome. High level of LINE-1 ORF1p expression is a hallmark of human cancer, reported in more than

**Figure 4. STING inhibition of LINE-1 depends on its ER exit.**

(A) Illustration of wild-type (WT) STING and its mutants. (B) Western blots to detect phosphorylation of IRF3 induced by wild-type STING or its mutants. (C) CMV-L1-neo$^{RT}$ colony assay was performed in HeLa cells with STING or its mutants. G418-resistant cell colonies were scored. ORF1p level was determined with Western blots at 48 h post transfection. The results of three independent experiments are presented in the bar graph (mean ± SEM; paired t-test). Compared with vector control, a significant difference was observed of STING ($p < 0.001$), S358A ($p < 0.001$), and S366A ($p < 0.001$) mutants; and there was no significant difference with del153-173 ($p = 0.984$), RIR/AIA ($p = 0.999$), and 238/240AA ($p = 0.997$) mutants. ns non-significant; *$p < 0.05$; **$p < 0.01$; ***$p < 0.001$. (D) Western blots to detect γH2AX in HeLa cells co-transfected with CMV-L1-neo$^{RT}$ and STING wildtype or its mutants. (E) Subcellular localization of STING and ORF1p at ER (Calnexin), ERGIC (ERGIC53) and Golgi (GM130) in STING-expressing HeLa cell line with BFA (Brefeldin A) (2 μM) treatment for 20 h. ImageJ was used to analyze the colocalization. Scale bar, 10 μm. Indicated regions are displayed enlarged. Histograms show the spatial distribution of the signal intensities across the sections indicated by red lines. The Trainable Weka Segmentation plugin in ImageJ was used to identify ORF1p-STING colocalization regions and calculate their colocalization coefficients with organelle markers (calnexin, ERGIC53, GM130). Statistical analysis of Pearson's correlation coefficients. ($n = 50$ cells analyzed, mean ± SEM). Compared with the STING-EGFP HeLa cell line post L1 transfection, there was no significant difference in ER ($p = 0.831$) and ERGIC ($p = 0.227$) colocalization coefficients, but a significant difference was observed in Golgi ($p < 0.001$) colocalization coefficients with BFA treatment. (F) Western blots to measure ORF1 protein level in HeLa cells co-transfected with STING and CMV-L1-neo$^{RT}$ DNA for 48 h, followed by treatment with BFA for 20 h. Protein bands in Western blots were quantified using ImageJ. The results of three independent experiments are presented in the bar graph (mean ± SEM; paired t-test). In DMSO treatment group, there were significant differences at 20 ng ($p < 0.001$), 40 ng ($p < 0.001$), 80 ng ($p < 0.001$), 160 ng ($p < 0.001$) and 320 ng ($p < 0.001$); in BFA treatment group, no significant difference was observed at 20 ng ($p = 0.984$), 40 ng ($p = 0.482$), 80 ng ($p = 0.494$), 160 ng ($p < 0.614$), and 320 ng ($p < 0.058$) compared with 0 ng STING-Flag transfection. ns non-significant; *$p < 0.05$; **$p < 0.01$; ***$p < 0.001$. Source data are available online for this figure.

90% of breast and ovarian cancers, nearly 90% of pancreatic cancers, 50–60% of tubular gastrointestinal tract cancers, 50% of lung cancers and 40% of prostate cancers (Rodic et al, 2014). Active LINE-1 retrotransposition can lead to chromosome instability (CIN), because LINE-1 ORF2p needs to cleave cellular DNA to initiate reverse transcription, and this DNA cleavage event triggers DNA damage response and CIN. High-level CIN is associated with poor prognosis of cancer patients, therapy resistance, immune evasion, tumor aggressiveness, and metastasis (Bakhoum and Cantley, 2018). We show that the number of γH2AX foci as a result of the expression of LINE-1 reporter system markedly increases in STING knockout cells, this suggests that STING suppresses the formation of double-strand DNA breaks associated with LINE-1 retrotransposition and DNA damage responses.

Previous studies have also shown that STING overexpression increases the association of DNA-PK complex proteins with the chromatin-nuclear matrix fraction and promotes cancer cell survival with chemotherapy treatment (Cheradame et al, 2021). Moreover, STING can contribute to genome stability in the steady state by regulating the formation of 53BP1 foci (Cheradame et al, 2021). The sources of genomic damage are diverse, and LINE-1 retrotransposon activity is merely one contributing factor. In addition, STING exhibits repair activity against genotoxic stressors such as chemotherapy-induced DNA damage in cancer cells, and influences tumor relapse and progression. In cell-based experiments without an overexpressed LINE-1 reporter system, both STING knockout and overexpression have been shown to modulate the accumulation of γH2AX foci number in the nucleus (Figs. 2K,L and EV2B,C). This indicates that STING regulates genomic stability not only by inhibiting LINE-1 transposition, but also by other mechanisms that affect genomic damage repair.

Active LINE-1 has also been associated with autoimmune diseases. For example, autoantibodies against LINE-1 p40 (ORF1p) have been reported in a group of SLE patients with severe and active disease (Carter et al, 2020). Hypomethylation of LINE-1 was observed in CD4 + T lymphocytes, CD8 + T lymphocytes, and B lymphocytes of SLE patients (Nakkuntod et al, 2011). It is postulated that LINE-1 DNA and RNA may serve as ligands of innate immune sensors, and trigger persistent production of interferons and inflammatory cytokines, thus contributing to autoimmune diseases such as

Aicardi–Goutières syndrome (AGS) and SLE. In this context, STING may be one cellular mechanism of preventing autoimmune diseases through controlling LINE-1 activity. In support of this role of STING, STING mutations R281Q and R284G were reported to associate with a devastating pediatric autoinflammatory disease SAVI (Melki et al, 2017). The SAVI pathology is independent of the type I IFN receptor (IFNAR1/2) and IFN-regulatory factors (IRFs), including IRF3 and IRF7, which are downstream of STING(Bennion et al, 2020; Luksch et al, 2019; Warner et al, 2017). The mutations R281Q and R284G are located in the polymerization interface and disturb oligomerization (Ergun et al, 2019; Shang et al, 2019). It is noted that R281 and R284 form one continuous motif with E282-D283, and our data demonstrate that the E282A/D283A mutant lost the ability of inhibiting LINE-1. It is possible that the contribution of R281Q and R284G to SAVI may be a result of the inability of these two STING mutants to control LINE-1 activity.

We observed changes in L1 mRNA levels in stable cell lines with STING overexpression or knockout, but not under transient overexpression or knockdown, as opposed to changes in ORF1p expression in both transiently transfected and stable cell lines. It is known that LINE-1 RNA binds to ORF1p and is thus protected from degradation. We postulate that in stable cell lines overexpressing STING, ORF1p is depleted for a prolonged time, which may have led to instability of LINE-1 RNA and its degradation and reduction. Under transient expression conditions, STING-mediated degradation of ORF1p may not have enough time to elicit a pronounced effect on LINE-1 RNA.

We found that STING inhibition of LINE-1 is independent of its function in stimulating IFN responses. STING is known for its response to cGAS sensing of DNA ligands. STING can bind to cGAMP synthesized by activated cGAS and cause TBK1 and IRF3 phosphorylation via its CTT. Our results showed that CTT is not required by STING to inhibit LINE-1, nor depletion of TBK1 has any effect on this activity of STING. Since STING homolog is present in almost all life forms including bacteria, and CTT is only present in STING homologs in vertebrates, it is possible that inhibiting transposons is an evolutionarily conserved function of STING. This adds to the growing list of STING functions independent of CTT and IFN signaling, such as NF-κB signaling, autophagy, calcium signaling, unfolded protein response (UPR),

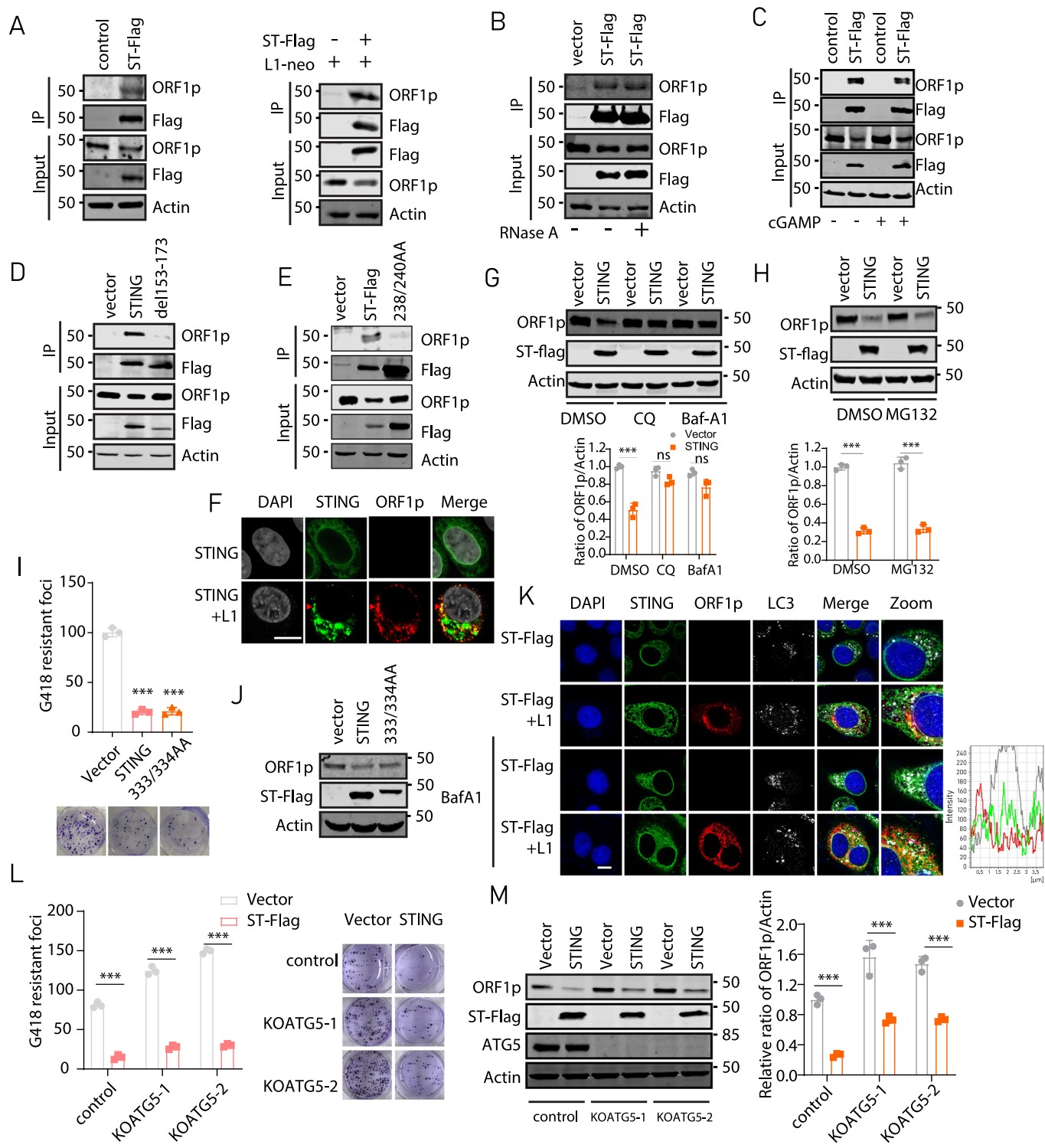

cell proliferation, and cellular senescence. For example, Xu lab reported that STING-PERK-elF2α complex supports an inflammatory- and survival-preferred translation program (Zhang et al, 2022). Recent findings reveal that STING also functions as a proton channel that deacidifies the Golgi apparatus and impacts Golgi enzyme activity, protein maturation, cargo sorting and secretion (Liu et al, 2023; Poddar et al, 2025; Xun et al, 2024). This pathway is

highly conserved and exists much earlier than the IFN pathway. Given the importance of guarding the stability and integrity of host genomes, controlling the activity of transposons may be an ancient function of STING homologs that has been preserved from non-vertebrates to vertebrates.

In addition, LINE-1 inhibition by STING is independent of cGAS but requires cGAMP binding site RY238/240. This is

**Figure 5. STING-mediated degradation of LINE-1 ORF1p is independent of autolysosomes.**

(A) Co-immunoprecipitation to detect the association of STING-Flag with endogenous ORF1p in HeLa cells stably-expressing STING-Flag (left). Co-immunoprecipitation to detect the association of STING-Flag with ORF1p in HEK293 cells co-transfected with CMV-L1-neo$^{RT}$ and STING-Flag DNA (right). (B) Effect of RNase A on co-immunoprecipitation of STING-Flag and ORF1p. HEK293 cells were co-transfected with CMV-L1-neo$^{RT}$ and STING-Flag. Co-IP was performed with or without RNase A. (C) Co-immunoprecipitation to detect the association of STING-Flag with ORF1p in HEK293 cells co-transfected with CMV-L1-neo$^{RT}$ and STING-Flag DNA with 6 h cGAMP treatment. (D) HEK293 cells were co-transfected with CMV-L1-neo$^{RT}$ and STING-Flag or del153-173 mutant. Co-IP was performed 48 h post transfection. (E) HEK293 cells were co-transfected with CMV-L1-neo$^{RT}$ and STING-Flag or 238/240AA mutant. Co-IP was performed 48 h post transfection. (F) Immunofluorescence of ORF1p in STING-EGFP-expressing HeLa cell line transfected with CMV-L1-neo$^{RT}$. Red triangle indicates colocalization. Scale bar, 10 μm. (G, H) Western blots to examine ORF1p degradation in HeLa cells co-transfected with CMV-L1-neo$^{RT}$ and STING-Flag, then treated with BafA1 (bafilomycin A1), CQ (chloroquine) (G) and MG132 (proteasome inhibitor) (H) for 6 h. Protein expression levels were quantified by scanning the Western blot bands and determining the grayscale intensity using ImageJ. The bar graph represents the relative density of the target protein normalized to actin. Data were presented as mean ± SEM from three independent experiments. Compared with the vector control, a significant difference was observed in DMSO ($p < 0.001$), but not in BafA1($p = 0.095$) and CQ ($p = 0.064$) groups (G). Compared with the vector control, significant differences were observed in the DMSO ($p < 0.001$) and MG132 ($p < 0.001$) group (H). Statistical significance was determined by a paired $t$-test. ns non-significant; *$p < 0.05$; **$p < 0.01$; ***$p < 0.001$. (I, J) Colony assay and Western blots were performed with HeLa cells co-transfected with CMV-L1-neo$^{RT}$ DNA and wild-type STING or the LR333/334AA mutant. The results of three independent experiments are presented in the bar graph (mean ± SEM; paired $t$-test). Compared with the vector control, significant differences were observed in STING-Flag ($p < 0.001$) and 333/334AA ($p < 0.001$). ns non-significant; *$p < 0.05$; **$p < 0.01$; ***$p < 0.001$. (K) Immunofluorescence was performed to detect the subcellular localization of ORF1p and LC3-GFP in the STING-Flag HeLa cell line with BafA1 treatment. STING in green, ORF1p in red, LC3 in gray. ImageJ was used for analyzing the colocalization. Scale bar, 10 μm. (L, M) Colony assay (L) and Western blots (M) were performed with ATG5 knockout HeLa cells transfected with CMV-L1-neo$^{RT}$ DNA. The results of three independent experiments are presented in the bar graph (mean ± SEM; paired $t$-test). Compared with the vector control, a significant difference was observed in STING-Flag transfection ($p < 0.001$), regardless of KOATG5-1/2 or Control cell lines in 5L and 5M. ns non-significant; *$p < 0.05$; **$p < 0.01$; ***$p < 0.001$. Source data are available online for this figure.

supported by the decreased expression of ORF1p in cells stably expressing the wild-type STING but not the RY238/240 mutant (Fig. EV3A,C). Similarly, LINE-1 activity is suppressed in cell lines expressing wild-type STING but not its mutant RY238/240AA (Fig. EV3B,D). These results rule out the effect of endogenous cGAMP and suggest that STING-mediated suppression of LINE-1 does not require the presence of cGAMP. The co-IP results demonstrated that the RY238/240AA mutation attenuated the STING-ORF1p interaction (Fig. 5E), indicating that these residues may disrupt the binding interface required for STING-ORF1p complex formation, while the precise underlying mechanism requires further investigation.

STING trafficking is conserved across evolution, even in single-cell organisms. STING is a transmembrane protein, exists on the ER with the CDN-binding domain facing the cytosol. Binding of CDN induces a structural rearrangement of STING, enables STING translocation to the ER-Golgi intermediate compartment (ERGIC) and the Golgi where it recruits the kinase TBK1 which phosphorylates STING and IRF3 which then initiates transcription of IFN-I genes and ISGs. Additionally, STING activates other IFN-independent signaling pathways during this process. ER to Golgi translocation of activated STING engages both autophagy and components of the trans-Golgi network. This convergently results in lysosomal degradation of STING, which is mediated by the endosomal sorting complex ESCRT associated with post-Golgi, clathrin-positive vesicles. Degradation of these vesicles by lysosomal compartments is defined as microautophagy but not conventional macroautophagy (Balka et al, 2023b; Gentili et al, 2023; Kuchitsu et al, 2023).

BFA treatment, which causes the collapse of the Golgi, partially restored ORF1p levels in the presence of STING (Fig. 4E,F), supporting a role of the canonical ER-to-Golgi trafficking pathway in ORF1p degradation by STING. In a previous study, the STING mutant LR333/334AA loses interaction with SEC24C upon cGAMP stimulation, thus cannot use COPII vesicles to exit ER (Gui et al, 2019). The interaction between ORF1 and STING might facilitate ER-to-Golgi transport through a non-canonical STING/SEC24C-

independent pathway. Alternatively, the classical ER-Golgi pathway may not be the exclusive mechanism for ORF1 degradation in lysosomes, as ER-phagy could also mediate ORF1p degradation (Liao et al, 2024; Sun et al, 2023).

The STING-ORF1p colocalized complex did not associate with LC3II, and STING-mediated inhibition of LINE-1 retrotransposition was unaffected upon depletion of key autophagy-related proteins (ULK1, ATG9a, Beclin1, p62, or ATG5). These results demonstrate that STING suppresses LINE-1 independently of both canonical and non-canonical macroautophagy. Furthermore, knockdown of core ESCRT components (TSG101 and ALIX) did not impair LINE-1 inhibition of STING, ruling out the involvement of ESCRT-dependent microautophagy. Surprisingly, individual knockout of TSG101 was found to impair the retrotransposition level of the LINE-1 reporter system, suggesting that TSG101 may play an auxiliary role in LINE-1 retro-transposition. The underlying mechanism remains unclear but appears to be independent of STING-mediated suppression of LINE-1 retrotransposition activity. Critically, the STING-ORF1p complex colocalized with the late endosome marker Rab7 and the lysosome marker LAMP1. These results collectively establish that STING targets ORF1p for degradation via the acidified endolysosomes.

In the GO BPs cluster analysis, STING knockout was also predominantly linked with regulation of vesicle-mediated transport in downregulated DEGs. Lysosomal degradation provides a negative feedback mechanism for controlling the activity of STING. Our study supports the model that STING may employ this inherent mechanism to clear harmful proteins such as LINE-1 ORF1p.

In summary, we report that STING has the function of suppressing the activity of LINE-1 through interacting with LINE-1 ORF1p and translocating to lysosomes for degradation. This function of STING is independent of its well-described role in regulating IFN signaling, may be conserved in its homologs in many species to protect host genomes from the assault by transposons.

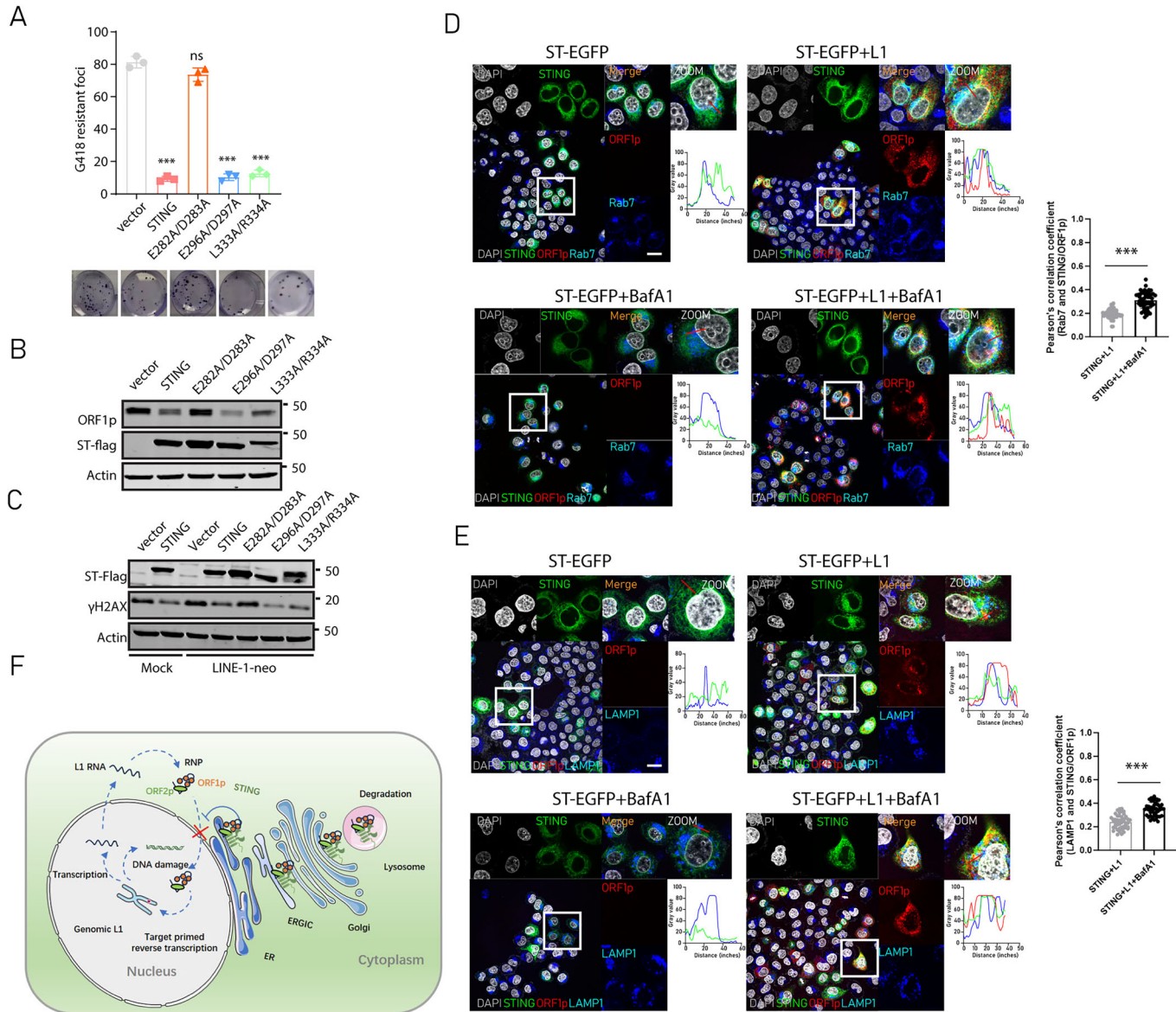

**Figure 6. STING inhibition of LINE-1 is dependent on endolysosomes.**

(**A**) Colony assay was performed to measure the effect of STING mutants E282A/D283A, E296A/D297A, and L333A/R334A on LINE-1 retrotransposition in HeLa cells. G418-resistant cell colonies were scored. The results of three independent experiments are presented in the bar graph (mean ± SEM; paired t-test). Compared with the vector control, a significant difference was observed in STING ($p < 0.001$), E282A/D283A ($p = 0.556$), E296A/D297A ($p < 0.001$) and L333A/R334A mutant ($p < 0.001$). ns non-significant; *$P < 0.05$; **$P < 0.01$; ***$P < 0.001$. (**B**) ORF1p protein level was determined by Western blots in HeLa cells co-transfected with STING wildtype or its mutants. (**C**) γH2AX level was determined with Western blots in HeLa cells co-transfected with CMV-L1-neo[RT] DNA and STING wildtype or its mutants. (**D**, **E**) Colocalization of STING-EGFP and ORF1p with Rab7 (**D**) or LAMP1 (**E**) in HeLa cells with BafA1 treatment. STING-EGFP-expressing HeLa cells were transfected with CMV-L1-neo[RT] for 24 h, followed by BafA1 treatment for 4 h. STING in Green, ORF1p in red, Rab7 in gray (**D**), LAMP1 in gray (**E**). Scale bar, 10 μm. Indicated regions are enlarged. Histograms show the spatial distribution of the signal intensities across the sections indicated by red lines. The Trainable Weka Segmentation plugin in ImageJ was used to identify ORF1p-STING colocalization regions and calculate their colocalization coefficients with organelle markers (Rab7 or LAMP1). Statistical analysis of Pearson's correlation coefficients. ($n = 50$ cells analyzed, mean ± SEM, paired t-test). Compared with the STING-EGFP HeLa cell line post L1 transfection, significant differences were observed in Rab7 ($p < 0.001$) and LAMP1 ($p < 0.001$) colocalization coefficients with BafA1 treatment. ns non-significant; *$p < 0.05$; **$p < 0.01$; ***$p < 0.001$. (**F**) Illustration of LINE-1 inhibition by STING. LINE-1 ORF1p and ORF2p associate with LINE-1 RNA and form the RNP complex. STING at the ER interacts with ORF1p and translocates LINE-1 RNP through the Golgi to lysosomes for degradation. Source data are available online for this figure.

# Methods

### Reagents and tools table

| Reagent/resource | Reference or source | Catalog number |
|---|---|---|
| **Experimental models** | | |
| HeLa | ATCC | CCL-2 |
| HEK293 | ATCC | 1573 |
| THP1 | ATCC | TIB-202 |
| **Antibodies** | | |
| Mouse anti-Actin antibody | Proteintech | 66009-1-Ig |
| Rabbit anti-STING antibody | Proteintech | 19851-1-AP |
| Rabbit anti-IRF3 antibody | Proteintech | 11312-1-AP |
| Rabbit anti-rab7 antibody | Proteintech | 55469-1-AP |
| Rabbit anti-ULK1 antibody | Proteintech | 20986-1-AP |
| Rabbit anti-ATG9a antibody | Proteintech | 26276-1-AP |
| Rabbit anti-P62 antibody | Proteintech | 18420-1-AP |
| Rabbit anti-Beclin1 antibody | Proteintech | 11306-1-AP |
| Rabbit anti-Alix antibody | Proteintech | 12422-1-AP |
| Mouse anti-TSG101 antibody | Proteintech | 67381-1-IG |
| Mouse anti-Flag antibody | Sigma | F1365 |
| Rabbit anti-ERGIC53 antibody | Sigma | E1031 |
| Mouse anti-LC3B antibody | Sigma | L7543 |
| Mouse anti-ORF1p antibody clone 4H1 | MILLIPORE | MABC1152 |
| Mouse anti-γH2AX antibody | MILLIPORE | 05–636 |
| Rabbit anti-calnexin antibody | Cell Signaling Technology | 2679 |
| Rabbit anti-TBK1 antibody | Cell Signaling Technology | 3013 |
| Rabbit anti-GM130 antibody | Cell Signaling Technology | 12480 |
| Rabbit anti-ATG5 antibody | Cell Signaling Technology | 12994 |
| Mouse anti-LAMP1 antibody | BioLegend | 328602 |
| Rabbit anti-phosphorylated IRF3 antibody | Abcam | ab76493 |
| Alexa Fluor 647 goat anti-rabbit antibody | Thermo Fisher Scientific | A21244 |
| Alexa Fluor 555 donkey anti-rabbit antibody | Thermo Fisher Scientific | A-21428 |
| Alexa Fluor 647-labeled donkey anti-mouse antibody | Thermo Fisher Scientific | A-21236 |
| **Chemicals, enzymes and other reagents** | | |
| DMEM medium | Thermo Fisher | C11995500BT |
| Fetal Bovine Serum | Gibco | A5669701 |
| Lipofectamine 2000 | Thermo Fisher | 11668030 |
| Lipofectamine RNAiMAX | Thermo Fisher | 13778030 |
| anti-Flag M2 affinity gel | Sigma-Aldrich | A2220 |
| cell-counting kit-8 | MCE | HY-K0301 |
| qPCR Master Mix | NEB | M3003L |

| Reagent/resource | Reference or source | Catalog number |
|---|---|---|
| cGAMP | InvivoGen | tlrl-nacga23 |
| **Software** | | |
| GraphPad Prism | https://www.graphpad.com/ | |
| LI-COR Image Studio | https://www.licor.com/ | |
| ImageJ | https://imagej.net/ij/ | |

## Plasmids

The CMV-L1-neo^RT reporter DNA contains a complete human LINE-1 DNA and a neomycin resistance gene in the LINE-1 3′UTR region (Esnault et al, 2000). The ORF1-YFP DNA was cloned into the pCMV-tag3B expression vector. pGL3-5′-UTR-Luciferase (5UTR-Luc) reporter plasmid contains LINE-1 5′-UTR promoter and Firefly luciferase gene (Li et al, 2013). The STING-Flag DNA was inserted into the pcDNA4 expression vector (Invitrogen). The STING-Flag mutations RY238/240AA, ED281/282AA, ED296/297AA, LR333/334AA, S358A, S366A, 178-180RIR/AIA, del153-173, and 1-340 were generated by the PCR-based mutagenesis method. The wild-type and mutated STING-EGFP DNA sequences were cloned into pEGFP-N1.

## Cell lines and cell culture

Human cervical carcinoma cell line HeLa (ATCC, CCL-2), human embryonic kidney HEK293 cells (ATCC, 1573) were grown in the Dulbecco's modified Eagle's medium (DMEM) supplemented with 10% fetal bovine serum (Invitrogen), 100 U/ml penicillin, and 100 μg/ml streptomycin. STINGKO cells was generated with the CRISPR-Cas9 system. LentiCRISPRv2 (52961, Addgene) which carries STING-targeted single guide RNAs (sgRNAs) were transfected into HeLa cells. Following puromycin (0.8 μg/mL) selection, resistant single-cell clones were acquired by serial dilution in 96-well plates.

STING gRNA: 5′-TCCATCCATCCCGTGTCCCA-3′

## Cell colony assay

HeLa cells were co-transfected with CMV-L1-neo^RT reporter DNA and wildtype or mutated STING DNA in 6-well plates. Three days post transfection, G418 (750 ug/ml) was added in medium to select for G418-resistant cell colonies. After 12 days of selection, cells were fixed and stained with crystal violet. G418-resistant colonies were scored.

## Assessment of toxicity

To assess potential cell toxicity caused by STING expression, 20,000 HeLa cells were seeded in 96-well plates. The next day, cells were transfected with STING plasmid DNA. Twenty-four hours post transfection, 10 μl cell-counting kit-8 was added to each well. After 2 h, OD450 was acquired with a microplate reader.

## Quantification of LINE-1 cDNA by PCR

HeLa cells were co-transfected with CMV-L1-neo[RT] plasmid and STING-Flag. Seventy-two hours post transfection, total cellular DNA was extracted using the QIAamp DNA mini kit. Extracted DNA (250 ng) was used as a template for PCR with primers L1cDNAF/L1cDNAR. The first primer 5′-CAGTTCGGCTGGCG CGAGGCC-3′ crosses the exon/intron junction within the neomycin resistance gene such that only the spliced and reverse transcribed DNA can be amplified. β-globin DNA was amplified with primers β-globinF/β-globinR, the results were used to normalize the levels of LINE-1 DNA. PCR products were resolved by electrophoresis in 1% agarose gel.

L1cDNAF: 5′-CAGTTCGGCTGGCGCGAGGCC-3′
L1cDNAR: 5′-CAGTTCCGCCCATTCTCCG-3′
β-globinF: 5′-TATTGGTCTCCTTAAACCTGTCTTG-3′
β-globinR: 5′-CTGACACAACTGTGTTCACTAGC-3′

## Quantification of LINE-1 RNA by RT-qPCR

Endogenous LINE-1 RNA levels were determined in HeLa cells that either overexpressed STING or had STING knocked down or knocked out. LINE-1 RNA levels were determined by qPCR using the Luna Universal qPCR Master Mix with primer pairs 5′UTRF/5′UTRR, ORF1pF/ORF1pR, ORF2pF/ ORF2pR (Hamdorf et al, 2015). The GAPDH RNA was measured as the internal control.

5′UTRF: 5′-AGAGAGCAGTGGTTCTCCCAGCACG-3′
5′UTRR: 5′-CTGTTGGAATACCCTGCCGTGTGAG-3′
ORF1pF: 5′-GCAGTTCCTCACCAGCAACAGAACA-3′
ORF1pR: 5′-GCCTTTGGTTTGAATGTCCTCCCG-3′
ORF2pF: 5′-TATTCAGGAAACCCATCTCACGTGC-3′
ORF2pR: 5′-GGGTGCTCCTGTATTGGGTGCATAA-3′
GAPDH-F: 5′-GCAAATTCCATGGCACCGT-3′
GAPDH-R: 5′-GCAAATTCCATGGCACCGT-3′

## Gene silencing

The target gene was knocked down with siRNA. HeLa cells were transfected with siRNA oligos using Lipofectamine RNAiMAX prior to plasmid transfection in six-well plates. Seventy-two hours post transfection, knockdown efficiency was examined by Western blotting.

siSTING-1: 5′-GCCUCAUUGCCUACCAGGAdTdT-3′
siSTING-2: 5′-GGCUUUAGCCGGGAGGAUAdTdT-3′
siSTING-3: 5′-GGCAUCAAGGAUCGGGUUUdTdT-3′
siATG5-1: 5′-GGCAACCUGACCAGAAACAtt-3′
siATG5-2: 5′-CCUUUCAUUCAGAAGCUGUtt-3′
siULK1-1: 5′-CGGAACAGCUGGUGCUGUAdTdT-3′
siULK1-2: 5′-GGGACAAGCCCAUGGCGCUdTdT-3′
siATG9a-1: 5′-CACUCAUGCACUUUGCCAUdTdT-3′
siATG9a-2: 5′-CCAACAAGAUGGUGAACCAdTdT-3′
siP62-1: 5′-CGUCUACAGGUGAACUCCAdTdT -3′
siP62-2: 5′-CACUUCGGGUGGCCAGGAUdTdT-3′
siBeclin1-1: 5′-GUUUGGAGAUCUUAGAGCAdTdT-3′
siBeclin1-2: 5′-GACUUGUUCCUUACGGAAAdTdT-3′
siALIX-1: 5′-CCUGGAUAAUGAUGAAGGATdT-3′
siALIX-2: 5′-GCAGUGAGGUUGUAAAUGUTdT-3′
siTSG101-1: 5′-CCGUUUAGAUCAAGAAGUATdT-3′
siTSG101-2: 5′-CCUCCAGUCUUCUCUCGUCTdT-3′

## Immunoprecipitation and Western blotting

RIPA buffer (0.1% SDS, 1% Triton X-100, 1% sodium deoxycholate, 150 mM NaCl, 10 mM Tris [pH 7.5], 1 mM EDTA) was used to lyse cells and then centrifuged at 12,000 rpm for 10 min at 4 °C to clarify the cell lysates. One milligram of cell lysate was incubated with 50 μl of anti-Flag gel (A2220 Sigma) for 15 min at 25 °C. The beads were washed three times with RIPA buffer, and then incubated with 2X loading buffer at 95 °C for 5 min to elute the bound protein.

The eluted proteins were separated in 12% SDS-PAGE, and then transferred onto nitrocellulose membranes. The membranes were probed with the indicated primary antibodies, followed by IRDyeTM secondary antibodies (1:20,000). Protein bands were visualized with a LI-COR Odyssey instrument.

## Immunofluorescence microscopy

HeLa cells were seeded on coverslips before transfection with the indicated plasmid. Twenty-four hours post transfection, cells were washed with PBS (pH 7.2), and fixed with 4% paraformaldehyde for 15 min at room temperature, followed by a 10 min permeabilization with 0.1% TX-100 at room temperature. Cells were then blocked in 5% BSA (bovine serum albumin) in PBS. Primary antibody was then incubated with cells at room temperature for 2 h. Secondary antibody was added and incubated for 1 h at room temperature. Confocal images were acquired using a Leica TCS SP5 mounted on an inverted microscope (DMI6000; Leica Microsystems) with an oil immersion 63x/NA1.4 objective lens (HCX PL APO CS; Leica Microsystems).

## RNA sequencing analysis

For mRNA and LncRNA sequencing analysis, total RNA from each sample was isolated with TRIzol reagent according to the manufacturer's instructions. Strand-specific library was constructed using NEBNext® Poly (A) mRNA Magnetic Isolation Module, and sequenced using Illumina Novaseq 6000 by Gene Denovo Biotechnology (Guangzhou). For miRNA sequencing analysis, isolated total RNAs (size range 18–30 nt) were enriched by PAGE. 3′ and 5′ adapters were ligated to RNAs, and ligation products were reverse-transcribed by PCR amplification, enriched to generate a cDNA library, and sequenced as above. High-quality clean reads were further filtered by fastp (version 0.18.0). Using ensembl_release 110 as the reference genome, an index of the reference genome was built, and paired-end clean reads were mapped to the reference genome using HISAT (version 2.1.0) with "-rna-strandness RF" and other parameters set as a default. The mapped reads of each sample were assembled by using StringTie (version 1.3.4) in a reference-based approach. For each transcription region, a TPM (Transcripts Per Kilobase of exon model per Million mapped reads) value was calculated to quantify its expression abundance and variations, using RSEM software.

## Statistics

All experiments were performed three or more times independently under similar conditions. All data were plotted as mean values, with variation as SEM. Statistical significance was calculated by Student's

two-tailed *t*-test. *P* values of statistical significance are represented as: ***$p < 0.001$; **$p < 0.01$; *$p < 0.05$.

## Data availability

The RNAseq data in this study were deposited in the Genome Warehouse in the National Genomics Data Center and are accessible at https://bigd.big.ac.cn/gsa through BioProject accession number PRJCA022301.

The source data of this paper are collected in the following database record: biostudies:S-SCDT-10_1038-S44319-025-00551-0.

## Peer review information

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

## Acknowledgements

This study was supported by funds from the National Key R&D Program of China (2023YFC2307902), from the National Natural Science Foundation of China (82241075, 82271802, 82072288, and 32200718), from CAMS Innovation Fund for Medical Sciences (CIFMS 2021-I2M-1-037, CIFMS 2021-I2M-1-038, CIFMS 2022-I2M-1-021, and CIFMS 2023-I2M-2-001). Confocal experiments were conducted at the Core Facilities and Service Centers, NIPB, CAMS&PUMC. We thank Yan Xiao and Jing Sun for expert assistance with experiments and analysis.

## Author contributions

**Yu Huang**: Data curation; Formal analysis; Investigation; Writing—original draft; Writing—review and editing. **Fengwen Xu**: Conceptualization; Data curation; Funding acquisition; Methodology; Writing—review and editing. **Lingwa Wang**: Resources; Formal analysis; Methodology. **Shan Mei**: Resources; Formal analysis; Validation; Methodology. **Fei Zhao**: Software; Investigation; Visualization; Methodology. **Liming Wang**: Software; Formal analysis; Visualization. **Yu Xie**: Software; Methodology. **Liang Wei**: Software; Methodology. **Yamei Hu**: Visualization. **Zhao Gao**: Methodology; Writing—original draft. **Tiffany Xue**: Software; Methodology. **Jugao Fang**: Conceptualization; Funding acquisition; Writing—review and editing. **Fei Guo**: Conceptualization; Supervision; Funding acquisition.

Source data underlying figure panels in this paper may have individual authorship assigned. Where available, figure panel/source data authorship is listed in the following database record: biostudies:S-SCDT-10_1038-S44319-025-00551-0.

## Disclosure and competing interests statement

The authors declare no competing interests.

# Expanded View Figures

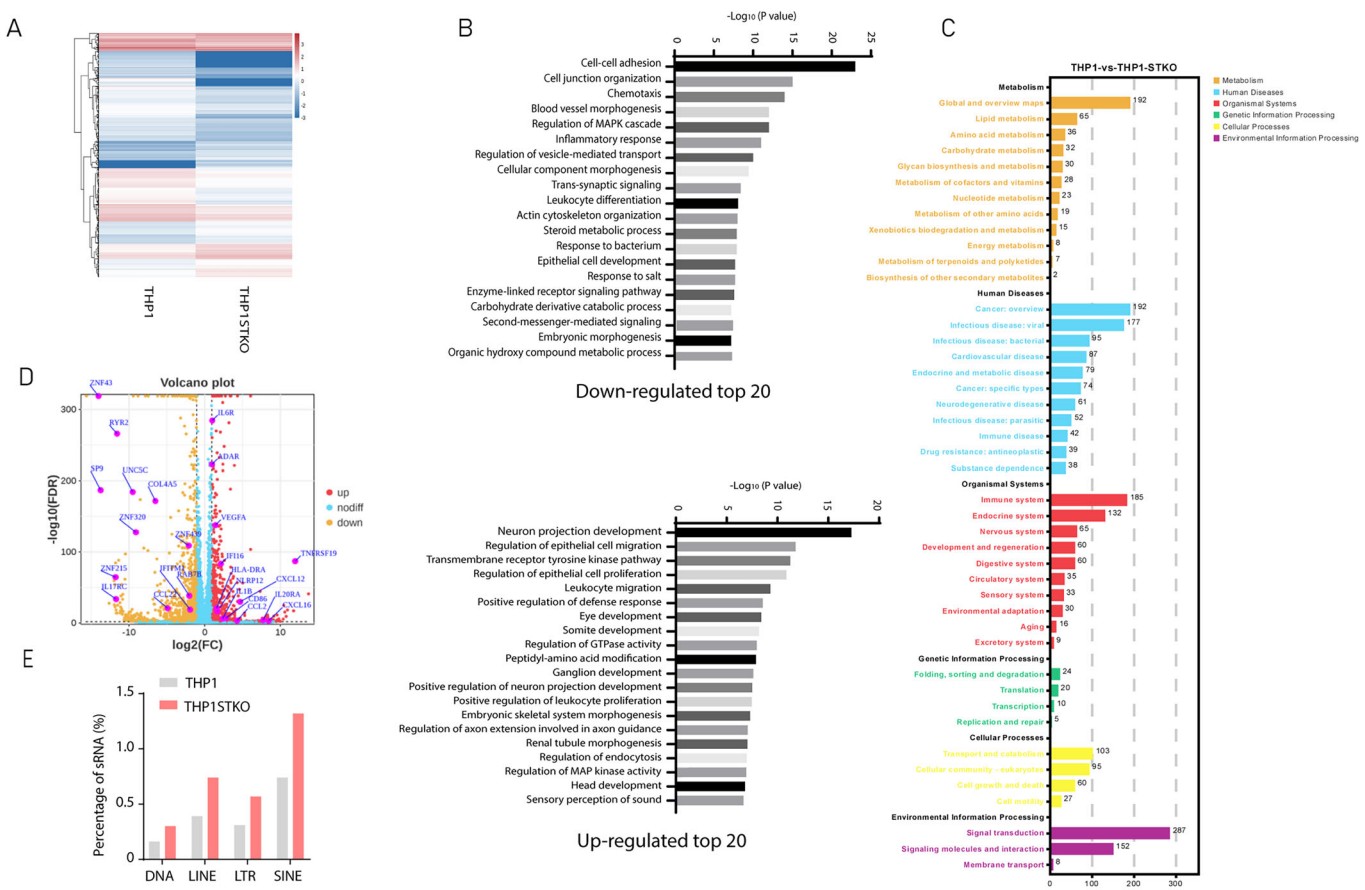

**Figure EV1. RNA sequencing analysis of DEGs and clustering between control and STING-knockout THP1 cells.**

(A) Expression heatmap of THP1 control and STING-knockout cells. (B) Bar graphs representing enriched top 20 GO BPs of upregulated and downregulated DEGs in STING-knockout THP1 cells with expression log2 [FC] >1 and *P* < 0.05, generated by Metascape. The *P* value was calculated based on the hypergeometric test. (C) Kyoto Encyclopedia of Genes and Genomes (KEGG) pathways of DEGs between control and STING-knockout THP1 cells. (D) Volcano plot of DEGs between control and STING knockout THP1 cells. The names of the DEGs associated with inflammation and immunity are shown. (E) Analysis of small RNA (18 to 30 nt) in STING-knockout THP1 cells.

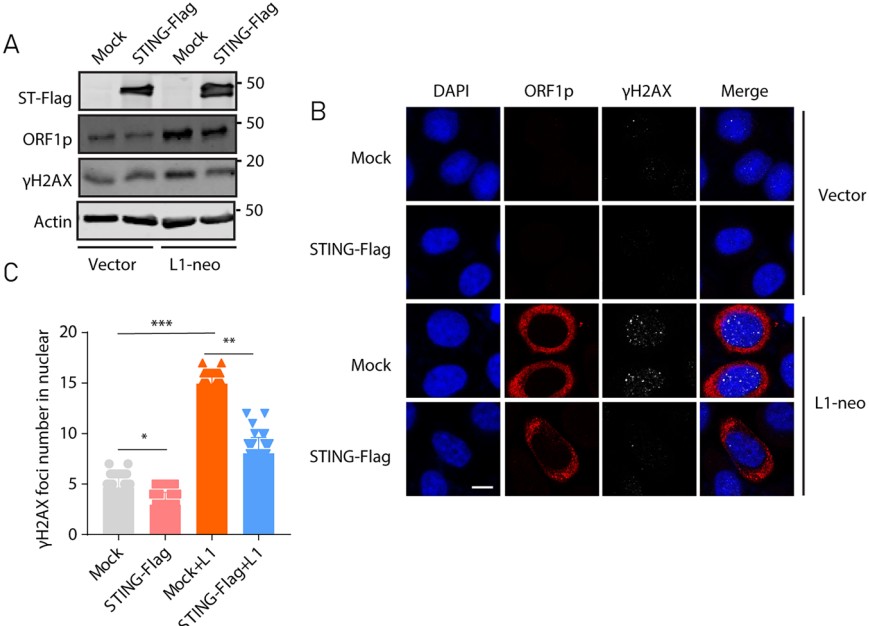

**Figure EV2. STING inhibits the formation of nuclear γH2AX foci induced by L1 retrotransposition.**

(A) Western blots to measure the levels of γH2AX in HeLa cells, which stably express STING-Flag and were transfected with the CMV-L1-neo$^{RT}$ DNA for 48 h. (B, C) Detection of γH2AX foci in HeLa cells which stably express STING-Flag and were transfected with the CMV-L1-neo$^{RT}$ DNA. Immunofluorescence was performed 24 h post transfection (B). Scale bar, 10 µm. γH2AX foci were scored in more than 50 cells for each treatment. The average number of γH2AX foci per cell is presented in the bar graph (C) (mean ± SEM; paired $t$-test). There was a significant difference ($p = 0.023$) in the STING-Flag stably-expressing cell line compared with the mock cell lines; a significant difference ($p < 0.001$) was also observed in the mock cell line with L1 transfection compared with the mock cell line; a significant difference ($p = 0.003$) was observed in STING-Flag stably-expressing cell line compared with the mock cell line which was transfected with L1 reporter system. ns non-significant; *$p < 0.05$; **$p < 0.01$; ***$p < 0.001$.

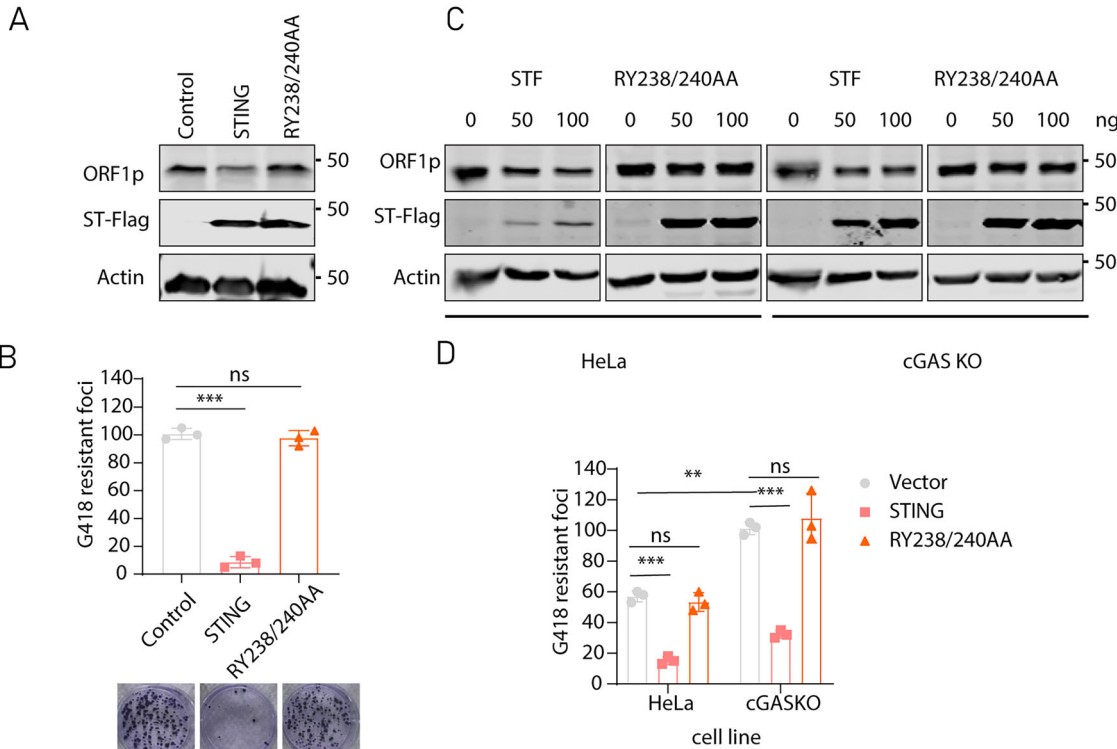

**Figure EV3.  STING RY238/240AA mutation impairs its ability to inhibit LINE-1 independently of cGAMP.**

(**A**) Levels of endogenous ORF1p were determined in HeLa cells stably expressing STING WT or RY238/240AA mutant. (**B**) CMV-L1-neo[RT] colony assay performed in HeLa cells stably expressing STING WT or RY238/240AA. The results of three independent experiments are presented in the bar graphs (mean ± SEM; paired *t*-test). Compared with the control group, a significant difference was observed in the STING-Flag stably-expressing cell lines ($p < 0.001$), but not in the RY238/240AA mutant stably-expressing cell lines ($p = 0.716$). ns, non-significant; *$p < 0.05$; **$p < 0.01$; ***$p < 0.001$. (**C**) STING WT or RY238/240AA mutant were co-transfected with CMV-L1-neo[RT] into HeLa or cGAS knockout cells. Levels of ORF1p were determined 48 h post transfection. (**D**) CMV-L1-neo[RT] colony assay performed with co-transfected STING WT or RY238/240AA mutant in cGAS knockout or control HeLa cells. The results of three independent experiments are presented in the bar graphs (mean ± SEM; paired *t*-test). In the control cell line, a significant difference was observed in the STING-Flag stably-expressing cell lines ($p < 0.001$), but not in the RY238/240AA mutant stably-expressing cell lines ($p = 0.807$); in the cGASKO cell line, there was also a significant difference in the STING-Flag stably-expressing cell lines ($p < 0.001$), but not in the RY238/240AA mutant stably-expressing cell lines ($p = 0.508$). ns non-significant; *$p < 0.05$; **$p < 0.01$; ***$p < 0.001$.

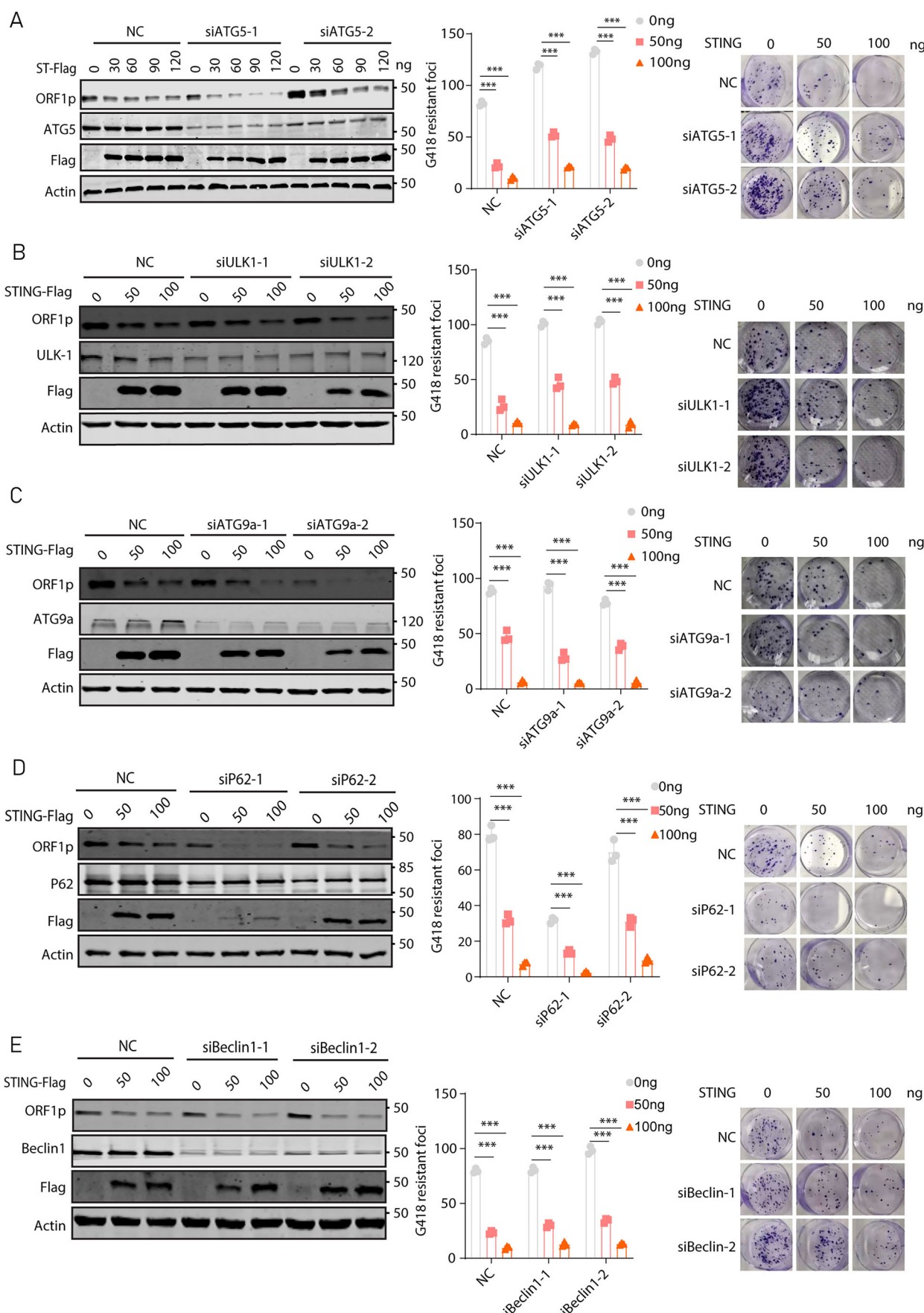

◄ **Figure EV4. Knockdown of autophagy factors does not affect STING inhibition of LINE-1.**

(A–E) ATG5 (**A**), ULK1 (**B**), ATG9a (**C**), P62 (**D**), or Beclin1 (**E**) was knocked down in HeLa cells with siRNA. followed by co-transfection with STING and CMV-L1-neo$^{RT}$ DNA. ORF1p was detected with Western blots at 48 h post transfection. A colony assay was performed to measure LINE-1 activity. The results of three independent experiments are presented in the bar graphs (mean ± SEM; paired *t*-test). Compared with 0 ng STING-Flag transfection, significant differences were observed at 50 ng ($p < 0.001$) and 100 ng ($p < 0.001$), independent of ATG5, ULK1, ATG9a, P62, or Beclin1 knockdown. ns non-significant; *$p < 0.05$; **$p < 0.01$; ***$p < 0.001$.

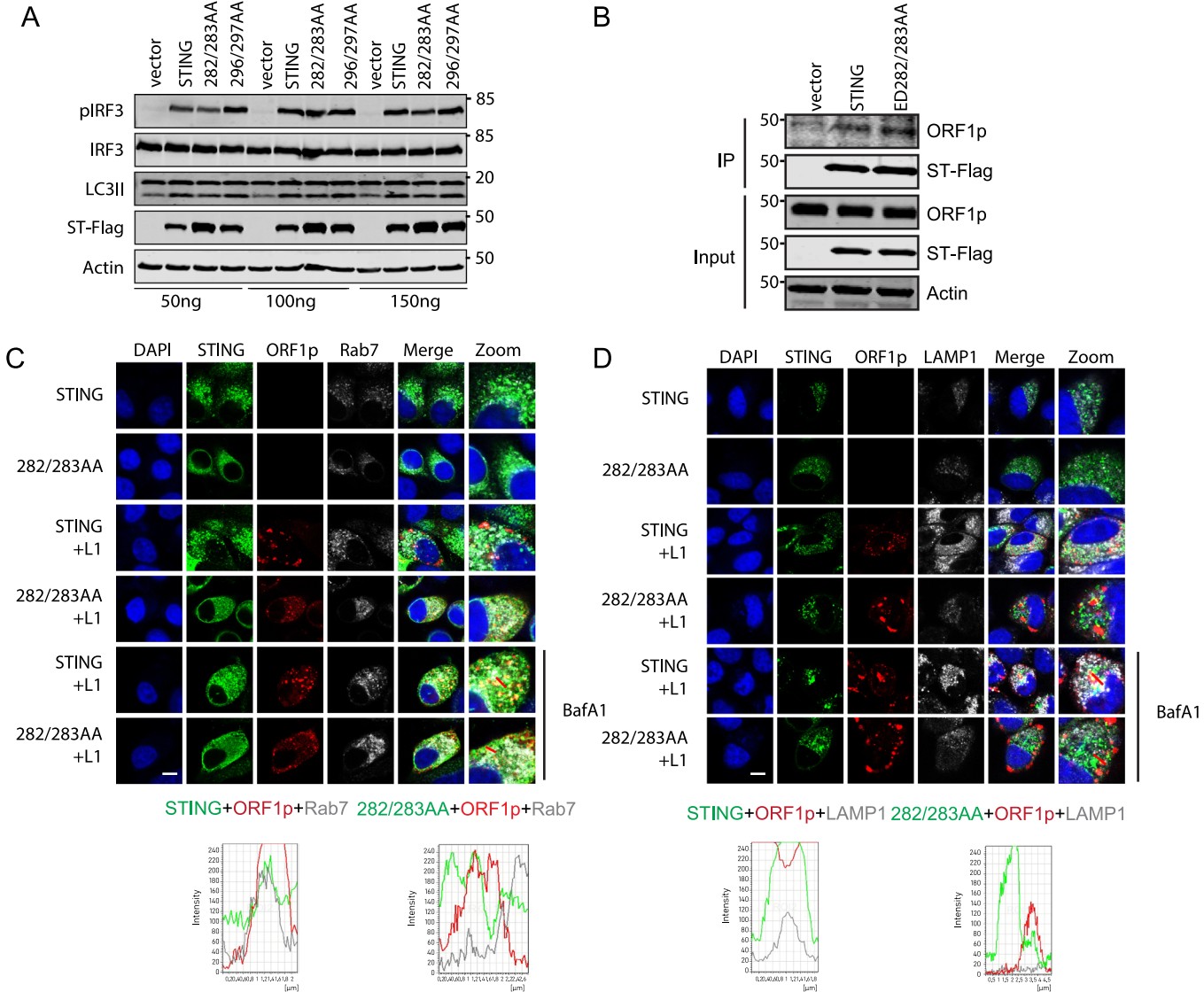

**Figure EV5. Amino acids E282/D283 of STING are required for trafficking of STING/ORF1p complex to lysosomes.**

(A) HeLa cells were transfected with STING and its mutants. Western blots were performed to detect IRF3 phosphorylation and LC3 lipidation. (B) HEK293 cells were co-transfected with CMV-L1-neo[RT] DNA and STING-Flag or its mutants. Immunoprecipitation was performed with anti-Flag antibody 48 h post transfection. Presence of ORF1p in the precipitated materials was detected by Western blots. (C) Colocalization of STING-EGFP or its E282A/D283A mutant with ORF1p and Rab7 in HeLa cells treated with BafA1. STING in Green, ORF1p in red, and Rab7 in gray. ImageJ was used to analyze the colocalization. Scale bar, 10 μm. (D) Colocalization of STING-EGFP or its E282A/D283A mutant with ORF1p and LAMP1 in HeLa cells treated with BafA1. STING in Green, ORF1p in red, and LAMP1 in gray. Red Line indicates colocalization. ImageJ was used to analyze the colocalization. Scale bar, 10 μm.

