## [Peer Review File · EMBO Reports]

STING inhibits LINE-1 retrotransposition through sorting ORF1p to lysosomes for degradation

Yu Huang, Fengwen Xu, Lingwa Wang, Shan Mei, Fei Zhao, Liming Wang, Yu Xie, Liang Wei, Yamei Hu, Zhao Gao, Tiffany Xue, Jugao Fang, and Fei Guo

Corresponding author(s): Fei Guo (guofei@ipb.pumc.edu.cn), Jugao Fang (fangjugao2@ccmu.edu.cn)

Review Timeline:

Submission Date:	5th Nov 24
Editorial Decision:	4th Dec 24
Revision Received:	3rd Mar 25
Editorial Decision:	3rd Apr 25
Revision Received:	27th Jun 25
Editorial Decision:	11th Jul 25
Revision Received:	17th Jul 25
Accepted:	29th Jul 25

Editor: Esther Schnapp

Transaction Report:

Dear Prof. Guo,

Thank you for the submission of your manuscript to EMBO reports. We have now received the comments from 2 referees and since both are in fair agreement, I am making a decision now in order to save time.

As you will see, the referees acknowledge that the findings are potentially interesting. However, both referees point out that the data are not sufficiently strong to support the main conclusions. It is clear that rather extensive revisions will be required before the study can be considered for publication here. Basically, all referee concerns will need to be addressed. In case you disagree, please let me know and we can discuss the revision requirements further, also in a video chat, if you like.

I would thus like to invite you to revise your manuscript with the understanding that the referee concerns must be fully addressed and their suggestions taken on board. Please address all referee concerns in a complete point-by-point response. Acceptance of the manuscript will depend on a positive outcome of a second round of review. It is EMBO reports policy to allow a single round of major revision only and acceptance or rejection of the manuscript will therefore depend on the completeness of your responses included in the next, final version of the manuscript.

We realize that it is difficult to revise to a specific deadline. In the interest of protecting the conceptual advance provided by the work, we recommend a revision within 3 months (6th Mar 2025). Please discuss the revision progress ahead of this time with the editor if you require more time to complete the revisions.

- 1) A data availability section providing access to data deposited in public databases is missing. If you have not deposited any data, please add a sentence to the data availability section that explains that.
- 2) Your manuscript contains statistics and error bars based on $n=2$. Please use scatter blots in these cases. No statistics should be calculated if $n=2$.

3) We replaced Supplementary Information with Expanded View (EV) Figures and Tables that are collapsible/expandable online. A maximum of 5 EV Figures can be typeset. EV Figures should be cited as 'Figure EV1, Figure EV2' etc... in the text and their respective legends should be included in the main text after the legends of regular figures.

5) a complete author checklist, which you can download from our author guidelines <https://www.embopress.org/page/journal/14693178/authorguide>. Please insert information in the checklist that is also reflected in the manuscript. The completed author checklist will also be part of the RPF.

6) Please note that all corresponding authors are required to supply an ORCID ID for their name upon submission of a revised manuscript (<https://orcid.org/>). Please find instructions on how to link your ORCID ID to your account in our manuscript tracking system in our Author guidelines <https://www.embopress.org/page/journal/14693178/authorguide#authorshipguidelines>

10) Regarding data quantification (see Figure Legends:

<https://www.embopress.org/page/journal/14693178/authorguide#figureformat>)

12) All Materials and Methods need to be described in the main text using our 'Structured Methods' format, which is required for all research articles. According to this format, the Methods section includes a separate Reagents and Tools Table file (listing key reagents, experimental models, software and relevant equipment and including their sources and relevant identifiers) followed by a Methods and Protocols section describing the methods using a step-by-step protocol format. The aim is to facilitate adoption of the methodologies across labs. More information on how to adhere to this format as well as a downloadable template (.docx) for the Reagents and Tools Table can be found in our author guidelines:

An example of a Method paper with Structured Methods can be found here: <https://www.embopress.org/doi/full/10.1038/s44320-024-00037-6#sec-4>

I look forward to seeing a revised form of your manuscript when it is ready.

Yours sincerely,

Referee #1:

This study by Huang et al. investigates the role of STING in retrotransposon suppression, specifically in the context of LINE-1 (L1) retrotransposition. The authors present evidence that STING downregulates both LINE-1 protein and mRNA levels, with the degradation of LINE-1 ORF1p appearing independent of STING-mediated TBK1 signaling. They propose that STING interacts with LINE-1 and facilitates its trafficking from the ER to the Golgi apparatus and, eventually, to lysosomes for degradation. This is a potentially novel function of STING with implications in aging and inflammation. However, although the effect of STING on the retrotransposition of L1-neoRT was dramatic, there are substantial concerns regarding the validity of the methods used throughout the study. Several important concerns need to be addressed to solidify the main conclusions.

Major Concerns:

1. Issues with Co-Transfection of CMV-L1-neoRT DNA and STING-Flag Plasmid: The co-transfection of CMV-L1-neoRT DNA and STING-Flag plasmids is used throughout the study without adequate controls. To resolve this issue, the authors should generate stable cell lines expressing L1-neoRT with or without STING, in order to fully rule out potential artifacts from transient transfection. If the cells express cGAS, transient transfection will induce cGAS activation, which could influence the results. Based on the data presented, transient transfection of STING appears to activate STING, which suggests that cGAS might be activated by the transfected plasmids. While STING mutants may serve as negative controls, the fact that these mutants still reduce ORF1p levels makes them inadequate as controls (quantification does not match western blot), as they cannot effectively distinguish between STING-specific effects and non-specific effects.
2. Dependence of STING's Role on cGAMP Binding: The authors claim that STING's function in ORF1p degradation is independent of cGAS but requires cGAMP (the product of cGAS). These claims appear contradictory. In cGAS knockout cells, where cGAMP is absent, STING-dependent LINE1 degradation was still claimed, suggesting that cGAMP is not required for this process. However, the cGAMP-binding mutant of STING was shown to be deficient in promoting LINE-1 degradation, which raises questions about the validity of these conclusions. To address these discrepancies, the authors should use stable cell lines to clarify the role of cGAMP in STING-dependent LINE-1 degradation.
3. Uncertainty in the Trafficking of STING and LINE1: The evidence for the trafficking of STING and LINE-1 is not clear. While some data in Figure 5C suggest LINE-1-dependent trafficking of STING, other images in the same condition (as shown in other panels) do not demonstrate STING puncta, making it difficult to draw reliable conclusions. Are these images representative at all? Moreover, there has been no careful quantification of whether STING and ORF1p traffic through the Golgi to lysosomes. The current quantification only presents data from a single punctum in one cell, which should not be the case. Large-scale colocalization quantification should be performed to assess trafficking more comprehensively. Additionally, the authors should avoid transient transfection when studying LC3 puncta and clarify the use of STING-EGFP versus STING-Flag in Figure 5H (the legend is not consistent with the figure).
4. Implications of STING Mutant LR333/334AA on ORF1p Degradation: Please note that the reason why STING LR333/334AA mutant is defective in autophagy is because it does not traffic from the ER upon cGAMP stimulation. However, it still promoted ORF1p degradation based on this study. Similarly, there was still substantial STING-dependent ORF1P degradation after Golgi disruption by BFA. These suggest that STING-mediated degradation might involve direct ER-phagy, independent of canonical ER-to-Golgi trafficking. However, macro-ER-phagy requires ATG5, but the authors also showed that ORF1p degradation does not rely on ATG5 (although not conclusive). This suggests that degradation may occur through micro-ER-phagy which does not require ATG5 or the Golgi. The authors could explore whether ESCRT-mediated ER-phagy is involved in ORF1p degradation (compare PMID 37922908 and 38593803). The authors should test ATG5 knockout cells to fully rule out potential autophagy-related effects.
5. Additional Clarifications for Future Studies: The authors should consider addressing the following basic questions about the process being studied:
 - o Does STING constitutively interact with LINE-1?
 - o Does cGAMP trigger STING binding to LINE-1?
 - o Does LINE-1 binding to STING trigger STING trafficking from the ER, independent of cGAMP?

Minor Comments:

1. Line1 mRNA Levels in STING-KO Cells: The upregulation of LINE-1 mRNA in STING-KO cells (Figure 1) needs further

explanation. If LINE-1 protein levels correlate with mRNA levels, why is there no change in LINE-1 mRNA observed in Figures 2C and D? Could this be due to the low levels of STING in HeLa cells compared to THP1 cells? The authors should consider knocking down or knockout STING in another cell type with higher endogenous STING expression to verify these results. Also, it would be useful to use Δ Ct calculations rather than absolute PCR values in Figure S2 to better assess relative mRNA expression.

2. Grammar: The manuscript contains grammatical errors that should be corrected for clarity and readability.
3. Cell Type Information: Please specify the cell type used in the legend of each figure panel for clarity.
4. Reference for Line 292: Please add the reference for the "previous report" mentioned in Line 292.
5. Figure 3E: The foci image in Figure 3E does not match the quantification; please verify.
6. Figure S6B: It appears that the wrong negative control was used in Figure S6B, or the labeling of STING and ORF1p is swapped.
7. Co-IP Control: add a negative control for the co-IP experiment between STING and ORF1p, such as using a STING mutant or a different ER-anchored protein.
8. Lysosomal Degradation of ORF1p: To confirm that ORF1p is degraded in lysosomes, the authors could express ORF1p-EGFP-mCherry to observe red-only signals upon lysosomal quenching of the EGFP signal.

Referee #2:

The authors disclosed a novel function of STING, which facilitates L1 ORF1 degradation to maintain genome integrity. They noted that cGAMP binding and STING dimerization are necessary, while cGAS is dispensable for this process. Furthermore, molecular dissection revealed that STING-mediated degradation of ORF1 relies on endolysosomes rather than autolysosomes. Despite the overall intriguing findings, several concerns hinder the manuscript's publication in its current state.

Major points

1. The manuscript lacks clarity on whether STING regulates L1 at the RNA level. In Figure 1, RNA-seq and RT-qPCR analyses in STING knockout THP1 cells indicated an increase in L1 RNA level. However, discrepancies in data shown in Figure 2 c-d and 2 f-g for HeLa cells is confusing. Although the luciferase assay was performed, it can represent only promoter activity, and may not sufficiently rule out the role of STING in regulating L1 at RNA-level.
2. The focus on post-translational regulation of STING overlooks potential RNA-level regulation. An associated concern is that while L1 RNA shows differential regulation in HeLa cells with transient STING overexpression/knockdown and in HeLa cells with STING stably overexpressed/knocked out, it is crucial for the authors to conduct experiments using cells with transient STING overexpression or knockdown to isolate the protein's impact. However, several experiments were performed using STING overexpression/knockout stable cell lines, where the observed alterations in STING levels may not solely stem from protein degradation.
3. STING participates in DNA repair (e.g. Cheradame et al., *Oncogene*, 2021). For Figure 2J-K, it is not clear that whether the effect of STING depletion on DNA damage accumulation is primarily dependent on its role on L1 regulation. The lack of a noticeable difference in γ H2AX foci fold change upon STING depletion with or without L1 overexpression does not provide substantial support for the conclusion "STING reduces γ H2AX foci induced by LINE-1 retrotransposition".
4. The authors discovered that cGAMP binding and dimerization are crucial for STING-mediated LINE-1 suppression. Interestingly, cGAS, a primary source of intracellular cGAMP, was not necessary for STING's role in regulating ORF1. This raises the question: What is the source of cGAMP in this particular context?
5. For Figure 4F, 5D and 5J, how many times were these experiments repeated? The authors should quantify these results.
6. The manuscript introduces the unexpected impact of STING depletion on inflammation but fails to explore the biological function of the STING-ORF1 axis. Does STING depletion promote inflammation through L1 activation? Are critical mutated residues in STING linked to inflammation-related diseases?

Minor points

1. All western blots should include molecular size markers.
2. For Figure S6 C-D, ensure the line indicating the region for colocalization analysis is clearly visible.
3. The language should be proofread by a native speaker. There are so many typos and grammar mistakes, such as "foci number in nuclear", "Figurers", "STING causes degradation of LINE-1 ORF1p is independent of autolysosomes".

Title: STING inhibits LINE-1 retrotransposition through sorting ORF1p to lysosomes for degradation

Manuscript number: EMBOR-2024-60709V1

We thank the editor and reviewers for their valuable comments, which have allowed us to substantially improve our manuscript.

Referee #1:

This study by Huang et al. investigates the role of STING in retrotransposon suppression, specifically in the context of LINE-1 (L1) retrotransposition. The authors present evidence that STING downregulates both LINE-1 protein and mRNA levels, with the degradation of LINE-1 ORF1p appearing independent of STING-mediated TBK1 signaling. They propose that STING interacts with LINE-1 and facilitates its trafficking from the ER to the Golgi apparatus and, eventually, to lysosomes for degradation. This is a potentially novel function of STING with implications in aging and inflammation. However, although the effect of STING on the retrotransposition of L1-neoRT was dramatic, there are substantial concerns regarding the validity of the methods used throughout the study. Several important concerns need to be addressed to solidify the main conclusions.

Major Concerns:

1. Issues with Co-Transfection of CMV-L1-neoRT DNA and STING-Flag Plasmid: The co-transfection of CMV-L1-neoRT DNA and STING-Flag plasmids is used throughout the study without adequate controls. To resolve this issue, the authors should generate stable cell lines expressing L1-neoRT with or without STING, in order to fully rule out potential artifacts from transient transfection. If the cells express cGAS, transient transfection will

induce cGAS activation, which could influence the results. Based on the data presented, transient transfection of STING appears to activate STING, which suggests that cGAS might be activated by the transfected plasmids. While STING mutants may serve as negative controls, the fact that these mutants still reduce ORF1p levels makes them inadequate as controls (quantification does not match western blot), as they cannot effectively distinguish between STING-specific effects and non-specific effects.

Response:

We agree with this reviewer on the possible activation of cGAS by the transiently transfected plasmid DNA. To rule out this possibility, we have performed the following experiments as suggested by the reviewer. 1) We have generated a HeLa cell line that stably expresses STING, and measured the levels of endogenous LINE-1 ORF1 protein and its mRNA, the results are described in lines 155 to 157, “in HeLa cell line stably expressing STING-Flag, endogenous ORF1 protein (Fig. 2E) and LINE-1 mRNA were decreased compared with control cell line (Fig. 2F), which is corroborated by the increases in the STING-knockout cell line (Fig. 2G and 2H).” 2) We performed RNA sequencing in the STING-knockout THP1 cell line to measure changes in the expression of LINE-1 RNA of different classes, the data are described in lines 120 to 121, “Further interrogation of RNAseq data showed an increase in the expression of 17 out of 26 classes full-length LINE-1 RNA in STING-knockout THP1 cells (Fig. 1A)”. 3) We performed G418 colony assay and measured ORF1p level in cGAS-knockout cells to rule out any effect of cGAS activation. The results are described in lines 184 to 189, “We transfected STING-Flag and CMV-L1-neo^{RT} plasmid to cGAS-knockout or control HeLa cells (V2) and scored the number of cell colony which reports LINE-1 activity. The results showed that STING remains inhibiting LINE-1 retrotransposition when cGAS was knocked out (Fig. 3A). ORF1 protein level was decreased in cGAS-knockout cells when STING was overexpressed (Fig. 3B). These results suggest that LINE-1 inhibition by STING is independent of cGAS.” 4) For the negative controls, we generated more STING mutants including ER retention/ retrieval sequence RIR (178-180aa), Δ 153-173 (defective in dimerization), RY238/240AA (defective in CDN binding), and E282A/D283A (defective in sorting to endolysosomes) (Fig. 6A and 6B). These STING mutants lost the ability of inhibiting LINE-1 retrotransposition and degrading ORF1p, serving as negative controls for the wild-type STING.

Since cell lines stably expressing CMV-L1-neo^{RT} will need G418 for selection, they cannot report CMV-L1-neo^{RT} activity anymore by scoring G418-resistance. We hope that results of the above four experiments are sufficient to rule out the possible involvement of cGAS in STING inhibition of LINE-1 activity.

2. Dependence of STING's Role on cGAMP Binding: The authors claim that STING's

function in ORF1p degradation is independent of cGAS but requires cGAMP (the product of cGAS). These claims appear contradictory. In cGAS knockout cells, where cGAMP is absent, STING-dependent LINE1 degradation was still claimed, suggesting that cGAMP is not required for this process. However, the cGAMP-binding mutant of STING was shown to be deficient in promoting LINE-1 degradation, which raises questions about the validity of these conclusions. To address these discrepancies, the authors should use stable cell lines to clarify the role of cGAMP in STING-dependent LINE-1 degradation.

Response:

We thank the reviewer and editor for highlighting this issue. As suggested by this reviewer, we have now generated a HeLa cell line stably expressing the STING RY238/240AA mutant and measured the effect on LINE-1 activity. The results are described in lines 230 to 236, “To avoid cGAMP production from plasmid DNA transfection, we generated a HeLa cell line that stably expresses STING RY238/240AA and assessed the expression of endogenous ORF1p. The results showed that the endogenous ORF1p expression level in the STING RY238/240AA cell line was comparable to that in the control cells (Fig. EV3A), indicating that the STING mutant RY238/240AA does not downregulate endogenous ORF1p. Additionally, LINE-1 activity from the transfected CMV-L1-neo^{RT} reporter system was not affected in the STING RY238/240AA stable cell line (Fig. EV3B).”

In lines of 236 to 240, “To further exclude cGAMP production, cGAS-knockout cells were utilized to overexpress STING along with the CMV-L1-neo^{RT} reporter system. Consistent with previous findings, STING significantly suppressed ORF1p expression and LINE-1 retrotransposition, while the RY238/240AA mutant lost this inhibition (Fig. EV3C and 3D). These results suggest that LINE-1 inhibition of STING requires the RY238/240 motif, not the presence of cGAMP.”

We have also discussed this point in lines 376 to 382, “In addition, LINE-1 inhibition by STING is independent of cGAS but requires cGAMP binding site RY238/240. This is supported by the decreased expression of ORF1p in cells stably expressing the wild type STING but not the RY238/240 mutant (Fig. EV3A and 3B). Similarly, LINE-1 activity is suppressed in cell lines expressing wild type STING wild-type but not its mutant RY238/240AA (Fig. EV3C and 3D). These results rule out the effect of endogenous cGAMP. Therefore, we conclude that STING-mediated suppression of LINE-1 does not require the presence of cGAMP, while the role of the STING motif RY238/240 in LINE-1 inhibition warrants further investigation.”

3. Uncertainty in the Trafficking of STING and LINE1: The evidence for the trafficking of

STING and LINE-1 is not clear. While some data in Figure 5C suggest LINE-1-dependent

trafficking of STING, other images in the same condition (as shown in other panels) do not demonstrate STING puncta, making it difficult to draw reliable conclusions. Are these images representative at all? Moreover, there has been no careful quantification of whether STING and ORF1p traffic through the Golgi to lysosomes. The current quantification only presents data from a single punctum in one cell, which is should not be the case. Large-scale colocalization quantification should be performed to assess trafficking more comprehensively. Additionally, the authors should avoid transient transfection when studying LC3 puncta and clarify the use of STING-EGFP versus STING-Flag in Figure 5H (the legend is not consistent with the figure).

Response:

We agree with this reviewer that STING punctates in the presence of LINE-1 ORF1p vary in their sizes between cells even under the same condition, this may reflect the dynamic nature of STING-ORF1p interaction, i.e. different numbers of these two proteins form assemblies of different sizes at different sites and at different times. To more accurately quantify protein co-localization and trafficking, we have marked multiple co-localized regions in Fig. 4E, and selected one of these regions in each panel for co-localization analysis using ImageJ. The degree of co-localization among the three fluorescence signals was represented by a curve. We described the results in lines 245 to 247, “ORF1p co-localized with STING in these membrane compartments (Fig. 4E), co-localization was marked with red triangles and presented with fluorescence intensity curve.”

We have now detected the puncta of endogenous CL3, but did not observe co-localization of LC3 with STING-EGFP/ORF1p, this result is presented in Fig. Appendix S5, and described in lines 277 to 281, “we performed immunofluorescence experiment to examine whether STING-induced LC3 puncta contain LINE-1 ORF1p, and did not observe co-localization of STING and ORF1p in LC3B-positive autophagosomes (GFP-labeled or endogenous LC3) with or without BafA1 treatment (Fig. 5G and Appendix Fig S5).” For the legend of original Fig. 5H (Fig. 5G now), STING-EGFP should be STING-Flag.

4. Implications of STING Mutant LR333/334AA on ORF1p Degradation: Please note that the reason why STING LR333/334AA mutant is defective in autophagy is because it does not traffic from the ER upon cGAMP stimulation. However, it still promoted ORF1p degradation based on this study. Similarly, there was still substantial STING-dependent

ORF1p degradation after Golgi disruption by BFA. These suggest that STING-mediated degradation might involve direct ER-phagy, independent of canonical ER-to-Golgi trafficking. However, macro-ER-phagy requires ATG5, but the authors also showed that ORF1p degradation does not rely on ATG5 (although not conclusive). This suggests that degradation may occur through micro-ER-phagy which does not require ATG5 or the Golgi. The authors could explore whether ESCRT-mediated ER-phagy is involved in ORF1p degradation (compare PMID 37922908 and 38593803). The authors should test ATG5 knockout cells to fully rule out potential autophagy-related effects.

Response:

As suggested by this reviewer, we have generated the ATG5-knockout HeLa cell line and examined STING inhibition of LINE-1. The results are described in lines 281 to 284, “We further knocked out ATG5 in HeLa cell to block STING-induced autophagy and measured the effect on CMV-L1-neo^{RT} retrotransposition activity. Results of LINE-1 reporter assay showed that STING inhibition of LINE-1 was not affected by ATG5 knockout (Fig. 5H). ATG5 knockout did not restore ORF1p expression (Fig. 5I).” The ATG5 knockout data are presented in Fig. 5 H and 5I, and ATG5 knockdown data are shown in Fig. EV4A.

In light that STING mutant LR333/334AA still inhibited LINE-1 and BFA treatment partially rescued STING inhibition of LINE-1, we agree with this reviewer that STING may employ two mechanisms to degrade ORF1p, either through ER-Golgi trafficking or ATG5-independent micro-ER-phagy. We have revised our conclusion in lines 249 to 250, “These results suggested that STING trafficking is crucial for the suppression of LINE-1 by STING”. We have also discussed this point in Discussion in lines 397 to 405, “The STING mutant LR333/334AA loses interaction with SEC24C, thus cannot use COPII vesicles to exit ER (Gui *et al.*, 2019). Since this mutant still stimulated ORF1p degradation, canonical ER-to-Golgi trafficking may not be the sole pathway underpinning STING-mediated ORF1p degradation. Instead, ER-phagy might have assisted STING to degrade ORF1p, a process involving the direct ripping off of a piece of the ER tubules by lysosomes (Liao *et al.*, 2024; Sun *et al.*, 2023). At the same time, BFA treatment, which causes collapse of Golgi, partially restored ORF1p levels in the presence of STING, supporting a role of the canonical secretory pathway in ORF1p degradation by STING. Therefore, both ER-Golgi trafficking and micro-ER-phagy might have contributed to ORF1p degradation by STING.”

5. Additional Clarifications for Future Studies: The authors should consider addressing the following basic questions about the process being studied:

o Does STING constitutively interact with LINE-1?

Response:

We have generated a HeLa cell line stably expressing STING-Flag, and performed co-immunoprecipitation (co-IP) experiments to detect the interaction between STING-Flag and endogenous LINE-1 ORF1p. The results are presented in Fig 5A, described in lines 257 to 258, “Indeed, the stably expressed STING-Flag was co-immunoprecipitated with endogenous ORF1p (Fig. 5A).” Data in the original Fig. 5A are shown in Appendix Figure S4A. These results suggest a constitutive interaction of STING with ORF1p.

o Does cGAMP trigger STING binding to LINE-1?

Response:

We have measured the interaction of STING-Flag and ORF1p with cGAMP treatment for 6 h, and did not observe any increase of the interaction compared to untreated cells. The results are presented in Appendix Figure S4C, described in lines 258 to 261, “Moreover, STING-Flag was co-immunoprecipitated with ORF1p that was expressed from CMV-L1-neo^{RT} (Appendix Fig. S4A), and this interaction was RNA- and cGAMP-independent (Appendix Fig. S4B and S4C), but required STING dimerization domain 153-173 (Appendix Fig. S4D)”

o Does LINE-1 binding to STING trigger STING trafficking from the ER, independent of

cGAMP?

Response:

We have addressed this question by examining STING trafficking in cGAS-knockout HeLa cells that were co-transfected STING-EGFP and CMV-L1-neo^{RT}. Co-localization of ORF1p and STING with the ER, ERGIC, and Golgi markers was observed (Appendix Fig. S3). We described these data in lines 251 to 254, “Furthermore, STING co-localized with ER, ERGIC and Golgi markers in cGAS-knockout cells that were co-transfected with STING-EGFP and CMV-L1-neo^{RT}, as shown by immunofluorescence analysis (Appendix Fig. S3). These findings further support that the trafficking of STING, upon its interaction with LINE-1 ORF1p, is independent of cGAS and cGAMP.”

Minor Comments:

1. Line1 mRNA Levels in STING-KO Cells: The upregulation of LINE-1 mRNA in

STING-KO cells (Figure 1) needs further explanation. If LINE-1 protein levels correlate

with mRNA levels, why is there no change in LINE-1 mRNA observed in Figures 2C and D?

Could this be due to the low levels of STING in HeLa cells compared to THP1 cells? The

authors should consider knocking down or knockout STING in another cell type with higher endogenous STING expression to verify these results. Also, it would be useful to use Δ Ct calculations rather than absolute PCR values in Figure S2 to better assess relative mRNA expression.

Response

Differential levels of LINE-1 mRNA in cells with transient expression of STING and stable expression of STING are now discussed in lines 354 to 361, “We observed changes in L1 mRNA levels in stable cell lines with STING overexpression or knockout, but not under transient overexpression or knockdown, as opposed to changes in ORF1p expression in both transiently transfected and stable cell lines. It is known that LINE-1 RNA binds to ORF1p and is thus protected from degradation. We postulate that in stable cell lines overexpressing STING, ORF1p is depleted for a prolonged time, which may have led to instability of LINE-1 RNA and its degradation and reduction. Under transient expression condition, STING-mediated degradation of ORF1p may not have long enough time to elicit pronounced effect on LINE-1 RNA.”

We have now knocked down STING in THP1 cells and measured endogenous ORF1p and its mRNA. The results showed a significant increase in endogenous ORF1p levels (Appendix Figure S2A), while no change in its mRNA (Appendix Figure S2B). We described these results in lines 153 to 155, “Knockdown of STING in THP1 cells similarly resulted in an upregulation of endogenous ORF1p (Appendix Fig. S2A), while its mRNA expression remained unchanged (Appendix Fig. S2B).”

Δ Ct has now been calculated to determine changes in LINE-1 cDNA levels. Figure S2 has now been changed to Appendix Figure S1.

2. Grammar: The manuscript contains grammatical errors that should be corrected for clarity and readability.

Response :

We have invited a native English speaker to proofread our manuscript and improve the clarity and readability.

3. Cell Type Information: Please specify the cell type used in the legend of each figure panel for clarity.

Response :

We have described each cell type in the figure legends to improve the clarity of manuscript.

4. Reference for Line 292: Please add the reference for the "previous report" mentioned in Line 292.

Response :

The reference has been added in line 310.

5. Figure 3E: The foci image in Figure 3E does not match the quantification; please verify.

Response:

We have now increased the brightness of the colony images to better visualize the big and small colonies. The quantification has been accurate.

6. Figure S6B: It appears that the wrong negative control was used in Figure S6B, or the labeling of STING and ORF1p is swapped.

Response:

We apologize for this mistake, the labeling of STING and ORF1p was swapped. This has been corrected in original Fig S6B (now Fig. EV5B).

7. Co-IP Control: add a negative control for the co-IP experiment between STING and ORF1p, such as using a STING mutant or a different ER-anchored protein.

Response:

We have now included a STING mutant del153-173 as the control. The results of co-IP showed that the del153-173 mutant of STING, which lacks dimerization capability, was significantly impaired in binding to ORF1p (Appendix Figure S4D).

8. Lysosomal Degradation of ORF1p: To confirm that ORF1p is degraded in lysosomes, the authors could express ORF1p-EGFP-mCherry to observe red-only signals upon lysosomal quenching of the EGFP signal.

Response:

We have generated an EGFP-mCherry-ORF1p plasmid, transfected into HeLa cells stably expressing STING-Flag, and performed confocal microscopy. The results are presented in Appendix Figure S6, described in lines 300 to 305, “To confirm that STING-mediated degradation of ORF1p takes place in lysosomes, we tagged ORF1p with both mCherry (pH insensitive sensor) and EGFP (pH-sensitive sensor), and transfected the plasmid into HeLa cells stably expressing STING-Flag. In the control cells, ORF1p was detected as yellow fluorescent signals. With expression of STING-Flag, ORF1p was detected mainly as red fluorescent signals with very weak green fluorescence, suggesting localization of ORF1p in lysosomes for degradation (Appendix Fig. S6).”

Referee #2:

The authors disclosed a novel function of STING, which facilitates L1 ORF1 degradation to maintain genome integrity. They noted that cGAMP binding and STING dimerization are necessary, while cGAS is dispensable for this process. Furthermore, molecular dissection revealed that STING-mediated degradation of ORF1 relies on endolysosomes rather than autolysosomes. Despite the overall intriguing findings, several concerns hinder the manuscript's publication in its current state.

Major points

1. The manuscript lacks clarity on whether STING regulates L1 at the RNA level. In Figure 1, RNA-seq and RT-qPCR analyses in STING knockout THP1 cells indicated an increase in L1 RNA level. However, discrepancies in data shown in Figure 2 c-d and 2 f-g for HeLa cells is confusing. Although the luciferase assay was performed, it can represent only promoter activity, and may not sufficiently rule out the role of STING in regulating L1 at RNA-level.

Response :

We have now discussed this discrepancy in lines 354 to 361, “We observed changes in L1 mRNA levels in stable cell lines with STING overexpression or knockout, but not under transient overexpression or knockdown, as opposed to changes in ORF1p expression in both transiently transfected and stable cell lines. It is known that LINE-1 RNA binds to ORF1p and is thus protected from degradation. We postulate that in stable cell lines overexpressing STING, ORF1p is depleted for a prolonged time, which may have led to instability of LINE-1 RNA and its degradation and reduction. Under transient expression condition, STING-mediated degradation of ORF1p may not have long enough time to elicit pronounced effect on LINE-1 RNA.”

2. The focus on post-translational regulation of STING overlooks potential RNA-level regulation. An associated concern is that while L1 RNA shows differential regulation in HeLa cells with transient STING overexpression/knockdown and in HeLa cells with STING stably overexpressed/knocked out, it is crucial for the authors to conduct experiments using cells with transient STING overexpression or knockdown to isolate the protein's impact. However, several experiments were performed using STING overexpression/knockout stable cell lines, where the observed alterations in STING levels may not solely stem from protein degradation.

Response :

As suggested by this reviewer, we have quantified the levels of LINE-1 ORF1p and its RNA levels with transient STING overexpression and transient knockdown, only levels of ORF1p protein, not its RNA, was affected (Fig. 2A 2B 2C and 2D). In addition, we also transfected the CMV-L1-neo^{RT} plasmid into cells stably expressing STING-Flag or with STING knockout, again ORF1p from this reporter LINE-1, not its RNA, was affected (Fig. R1). These data suggest that STING does not directly affect the expression of ORF1p RNA.

Fig R1. CMV-L1-neo^{RT} mRNA was unaffected in STING-Flag stably expressed or STING knockout cell line

(A and B) STING knockout HeLa cell line was transfected with 250 ng CMV-L1-neo^{RT} reporter plasmid, 48 hours post-transfection, western blot and RT-qPCR were performed to detect ORF1p and its mRNA (mean \pm SEM; paired *t*-test). ns, no significant.

(C and D) STING stably expressed HeLa cell line was transfected with 250 ng CMV-L1-neo^{RT} reporter plasmid, 48 hours post-transfection, western blot and RT-qPCR were performed to detect ORF1p and mRNA (mean \pm SEM; paired *t*-test). ns, no significant.

3. STING participates in DNA repair (e.g. Cheradame et al., Oncogene, 2021). For Figure 2J-K, it is not clear that whether the effect of STING depletion on DNA damage accumulation is primarily dependent on its role on L1 regulation. The lack of a noticeable difference in γ H2AX foci fold change upon STING depletion with or without L1 overexpression does not provide substantial support for the conclusion "STING reduces γ H2AX foci induced by LINE-1 retrotransposition".

Response:

We have now included the vector plasmid transfection as the control, and observed significantly higher levels of γ H2AX and its foci in STING KO cells transfected with CMV-L1-neo^{RT} than those transfected with vector plasmid (Fig. 2J-2L). We further discussed DNA repair function of STING in lines 331 to 336, "Previous studies have also shown that STING overexpression increases the association of DNA-PK complex proteins with the chromatin-nuclear matrix fraction and promotes cancer cell survival with chemotherapy treatment (Cheradame et al, 2021). Moreover, STING can contribute to genome stability in steady state by regulating the formation of 53BP1 foci (Cheradame et al., 2021). Our study suggests that STING may protect genomes from DNA damage by suppressing LINE-1 activity."

4. The authors discovered that cGAMP binding and dimerization are crucial for STING-mediated LINE-1 suppression. Interestingly, cGAS, a primary source of intracellular cGAMP, was not necessary for STING's role in regulating ORF1. This raises the question: What is the source of cGAMP in this particular context?

Response:

This question was also raised by the other reviewer, our answer is the following:

we have now generated a HeLa cell line stably expressing the STING RY238/240AA mutant and measured the effect on LINE-1 activity. The results are described in lines 230 to 236, “To avoid cGAMP production from plasmid DNA transfection, we generated a HeLa cell line that stably expresses STING RY238/240AA and assessed the expression of endogenous ORF1p. The results showed that the endogenous ORF1p expression level in the STING RY238/240AA cell line was comparable to that in the control cells (Fig. EV3A), indicating that the STING mutant RY238/240AA does not downregulate endogenous ORF1p. Additionally, LINE-1 activity from the transfected CMV-L1-neo^{RT} reporter system was not affected in the STING RY238/240AA stable cell line (Fig. EV3B).”

In lines of 236 to 240, “To further exclude cGAMP production, cGAS-knockout cells were utilized to overexpress STING along with the CMV-L1-neo^{RT} reporter system. Consistent with previous findings, STING significantly suppressed ORF1p expression and LINE-1 retrotransposition, while the RY238/240AA mutant lost this inhibition (Fig. EV3C and 3D). These results suggest that LINE-1 inhibition of STING requires the RY238/240 motif, not the presence of cGAMP.”

We have also discussed this point in lines 376 to 382, “In addition, LINE-1 inhibition by STING is independent of cGAS but requires cGAMP binding site RY238/240. This is supported by the decreased expression of ORF1p in cells stably expressing the wild type STING but not the RY238/240 mutant (Fig. EV3A and 3B). Similarly, LINE-1 activity is suppressed in cell lines expressing wild type STING wild-type but not its mutant RY238/240AA (Fig. EV3C and 3D). These results rule out the effect of endogenous cGAMP. Therefore, we conclude that STING-mediated suppression of LINE-1 does not require the presence of cGAMP, while the role of the STING motif RY238/240 in LINE-1 inhibition warrants further investigation.”

5. For Figure 4F, 5D and 5J, how many times were these experiments repeated? The authors should quantify these results.

Response:

These experiments were repeated at least three times. The signals from three experiments were quantified and summarized in the bar graphs. The original Fig 5J of siATG5 in HeLa cells has now become Fig. EV4A, and the data from ATG5 knockout HeLa cells line are presented in Fig. 5I (as suggested by reviewer 1).

6. The manuscript introduces the unexpected impact of STING depletion on inflammation but fails to explore the biological function of the STING-ORF1 axis. Does STING depletion promote inflammation through L1 activation? Are critical mutated residues in STING linked to inflammation-related diseases?

Response:

We have now discussed the possible role of LINE-1 activity in inflammation and autoimmune diseases caused by STING deficiency. In lines 58 to 62, “STING^{-/-} MRL.Fas^{lpr} mice exhibit more severe SLE disease and accelerated death. STING-knockout mice are hyperresponsive to TLR7/9 ligands including endogenous nucleic acids (Sharma *et al*, 2015a), which may include reactivated retro-elements that can stimulate IFN-I response and promote inflammation and autoimmunity (Mavragani *et al*, 2016; Sharma *et al*, 2015b).” In lines 337 to 345, “Active LINE-1 has also been associated with autoimmune diseases. For example, autoantibodies against LINE-1 p40 (ORF1p) have been reported in a group of SLE patients with severe and active disease(Carter *et al.*, 2020). Hypomethylation of LINE-1 was observed in CD4+ T lymphocytes, CD8+ T lymphocytes, and B lymphocytes of SLE patients(Nakkuntod *et al*, 2011). It is postulated that LINE-1 DNA and RNA may serve as ligands of innate immune sensors, and trigger persistent production of interferons and inflammatory cytokines, thus contribute to autoimmune diseases such as Aicardi–Goutières syndrome (AGS) and SLE. In this context, STING may be one cellular mechanism of preventing autoimmune diseases through controlling LINE-1 activity.”

We also discussed STING mutations in lines 345 to 353, “STING mutations R281Q and R284G were reported to associate with a devastating pediatric autoinflammatory disease SAVI(Melki *et al*, 2017). The SAVI pathology is independent of the type I IFN receptor (IFNAR1/2) and IFN-regulatory factors (IRFs) including IRF3 and IRF7, which are downstream of STING(Bennion *et al*, 2020; Luksch *et al*, 2019; Warner *et al*, 2017). The mutations R281Q and R284G are located in the polymerization interface and disturb oligomerization(Ergun *et al*, 2019; Shang *et al*, 2019). It is noted that R281 and R284 form one continuous motif with E282-D283, and our data demonstrate that the E282A/E283A mutant lost the ability of inhibiting LINE-1. It is possible that contribution of R281Q and R284G to SAVI may be a result of the inability of these two STING mutants to control LINE-1 activity.”

Minor points

1. All western blots should include molecular size markers.

Response:

Molecular size markers have been added to all western blots.

2. For Figure S6 C-D, ensure the line indicating the region for colocalization analysis is clearly visible.

Response

Line is now bold and changed to red to clearly highlight the region of colocalization.

3. The language should be proofread by a native speaker. There are so many typos and

grammar mistakes, such as "foci number in nuclear", "Figuers", "STING causes degradation of LINE-1 ORF1p is independent of autolysosomes".

Response:

We have invited a native English speaker to help us improve the manuscript. “Figuers” has been corrected to “Figures”. In line 255, “STING causes degradation of LINE-1 ORF1p is independent of autolysosomes” has been changed to “STING-mediated degradation of LINE-1 ORF1p is independent of autolysosome.”

Dear Prof. Guo,

Thank you for the submission of your revised manuscript. We have now received the comments from both referees as well as referee cross-comments that are all pasted below.

As you will see, while the referees acknowledge that the study has been improved, both of them also point out that additional revisions will be required before the work can be considered for publication here. In their cross-comments, the referees agree with each others concerns, and we have decided that all comments should be addressed. Please provide a point-by-point response with your final ms to all final comments.

Some editorial requests will also need to be addressed:

- Please remove the figures from the ms file.
- Please correct the conflict of interest subheading to "Disclosure and Competing Interests Statement"
- Jugao Fang is missing an institutional email address, which must be provided.
- The author credits need to be removed from the ms file. All credits need to be entered during online ms submission.
- The FUNDING INFO needs to be part of the Acknowledgments.
- The APPENDIX FILE is missing a title page with a table of content including page numbers for the Appendix items
- Please upload the Reagents & Tools table as a separate file.
- The new panel Source Data needs to be part of their corresponding SD Figure folder.
- Materials and Methods should be just "Methods"
- The Supplemental information section needs to be removed from the ms.
- Figure legends:
 1. Please note that the exact p-values are not provided in the legends of figures 1C, E, F, G, I, J, K, L; 2C, D, F, H, I, L; 3A, E, H, K; 4C, F; 5C, D, E, H, I; 6A, EV2 C, EV3 B, D; EV4 A-E. Please provide exact p-values as reasonable.
 2. Please indicate the statistical test used for data analysis in the legends of figures EV1 B, EV4 A-E
 3. Please note that information related to n is missing in the legends of figures EV3 B, D ; EV4 A-E
 4. Please note that the error bars are not defined in the legends of figures EV4 A-E

EMBO press papers are accompanied online by A) a short (1-2 sentences) summary of the findings and their significance, B) 2-3 bullet points highlighting key results and C) a synopsis image that is exactly 550 pixels wide and 200-600 pixels high (the height is variable). The synopsis image should provide a sketch of the major findings, like a graphical abstract. Please note that text needs to be readable at the final size. Please send us this information along with the final manuscript.

I would like to suggest some minor changes to the abstract. Do you agree with:

The cyclic dinucleotide sensor stimulator of interferon (IFN) genes (STING) is known for its critical role in interferon and inflammatory responses. In addition, STING also has functions independent of interferon induction. In this study, we report that STING restricts the mobilization of the cellular retrotransposon long interspersed nuclear element 1 (LINE-1) independent of cGAS and interferon induction. LINE-1 is the only active autonomous retrotransposable element in the human genome and its transposition can cause genetic and autoimmune diseases. STING inhibition of LINE-1 requires its dimerization. Mechanistically, STING interacts with LINE-1 ORF1p, then the complex translocates to the ER-Golgi intermediate compartment (ERGIC) and the Golgi followed by sorting to Rab7-positive lysosomes for degradation. Our data unveil a function of STING in maintaining host genome integrity by restricting LINE-1 retrotransposition via an IFN-independent mechanism.

I look forward to seeing a final version of your manuscript when it is ready.

Kind regards,

Esther

Referee #1:

I appreciate the authors' effort in revising the manuscript. However, the study now concludes without a clear explanation of how LINE-1 is degraded by STING. Additionally, there is a surprising requirement for the cGAMP-binding pocket, despite cGAMP having no impact. I recommend another revision to address the following two questions, which, if answered, would significantly improve the impact of the paper.

1. The claim that "STING trafficking is crucial for the suppression of LINE-1 by STING" is in question. The STING trafficking-deficient mutant LR333/334AA is as potent as STING-WT in ORP1p regulation (Fig 5F, 6B), suggesting that the regulation is independent of STING trafficking from the ER to the Golgi. ESCRT knockdown (siALIX+siTSG101, see PMID 29622626) will be a quick test to find out if microautophagy is essential.

2. Line 376, "In addition, LINE-1 inhibition by STING is independent of cGAS but requires cGAMP binding site RY238/240". This is still rather surprising, unless the cGAMP-binding pocket also binds to LINE-1. Please assess if the cGAMP-binding mutant of STING RY238/240AA still associates with LINE-1 in co-IP. This would be a better control than the dimerization mutant (Appendix Fig. S4D) for the co-IP experiment as without dimerization, the protein is likely misfolded.

Minor:

1. The only evidence to support a requirement of Golgi trafficking was BFA treatment which is quite toxic. It is thus over interpretation to claim that "STING trafficking is crucial for the suppression of LINE-1 by STING". Major point 1, if works, would answer the mechanism of LINE1 degradation by STING.
2. Figure 4E still has low image quality and lacks statistic support. As the requirement for STING trafficking in LINE1 degradation is weak, please either remove the whole panel or improve it (add colocalization quantification from 50-100 cells/condition). Similar for Fig 6D/E to ensure reproducibility.
3. lines 230 to 236, "To avoid cGAMP production from plasmid DNA transfection, we generated a HeLa cell line that stably expresses STING RY238/240AA...". Note that the cGAMP-binding deficient STING mutant cannot avoid cGAMP production by cGAS, but it can avoid the impact of cGAMP to STING. please reword accordingly.

Referee #2:

The authors have addressed some of my concerns, but several critical issues remain unresolved. These concerns must be carefully addressed before the manuscript can be considered for publication.

1. There is an inconsistency in the γ H2AX levels presented in Figures 2J-K. Western blotting shows no significant change in γ H2AX following L1 overexpression, whereas immunofluorescence indicates a marked upregulation. How can this discrepancy be explained? Additionally, it would be beneficial to provide quantification for Figure 2J.
2. The discussion regarding STING's role in DNA repair is not accurate. A previous study has demonstrated that STING promotes DNA repair, suggesting that the impact on γ H2AX levels in STING KO cells may not be solely attributed to its regulation on L1. This is further supported by the observation of increased γ H2AX levels in STING KO cells in the absence of L1 overexpression, as shown in Figure 2J.
3. In the first round of review, both Reviewer 1 and I raised concerns about the claimed cGAS-independent but cGAMP-mediated function in ORF1p degradation. Given that cGAS is responsible for the synthesis of cGAMP and cGAS is dispensable for ORF1p degradation, why is cGAMP-mediated oligomerization of STING considered indispensable? The authors have yet to provide compelling experimental evidence or discussion on this contradictory point.

Cross-comments from referee 1:

I agreed with all the points (only 3 points) from Reviewer 2, who I believe was also trying to help based on the comments. I have already substantially lowered my standards when asking for more data in the 2nd round. Based on Reviewer 2's comments, I think it is the same case. The authors added a lot of work but failed to address the most important questions. I am a little reluctant to reject to it, so I suggest a 2nd chance. I always try to be as supportive and constructive as possible when reviewing, with clear experimental suggestions. I hope the authors find the new comments helpful.

Cross-comments from referee 2:

I agree with the other reviewer's point. They should be addressed.

Response to comments on EMBO Reports submission EMBOR-2024-60709V2. “STING inhibits LINE-1 retrotransposition through sorting ORF1p to lysosomes for degradation.”

We thank the editor and reviewers for their valuable comments, which allowed us to substantially improve our manuscript. We now provide a point-by-point response detailing all additional experiments and revisions included in this manuscript, as outlined below.

Editorial comments:

- Please remove the figures from the ms file.

Response:

Figures have been removed from the ms file.

- Please correct the conflict of interest subheading to “Disclosure and Competing Interests Statement”

Response:

Conflict of interest has been corrected to “Disclosure and Competing Interests Statement”.

- Jugao Fang is missing an institutional email address, which must be provided.

Response:

Institutional email address (fangjugao2@ccmu.edu.cn) has been provided in ms file.

- The author credits need to be removed from the ms file. All credits need to be entered during online ms submission.

Response:

The author credits have been moved and fill out during online submission.

- The FUNDING INFO needs to be part of the Acknowledgments.

Response:

Funding information have been integrated into a part of Acknowledgments.

- The APPENDIX FILE is missing a title page with a table of content including page numbers for the Appendix items

Response:

A title page with a table of content have been supplemented with APPENDIX FILE.

- Please upload the Reagents & Tools table as a separate file.

Response:

A separate Reagents & Tools table file has been provided online.

- The new panel Source Data needs to be part of their corresponding SD Figure folder.

Response:

New panel Source Data of Fig 2J, 4E, 6D, 6E, Appendix Figure S4E and S7 were provided as a part of corresponding SD Figure folder.

- Materials and Methods should be just “Methods”

Response:

“Materials and Methods” has been corrected with “Methods”.

- The Supplemental information section needs to be removed from the ms.

Response:

Supplemental information section has been removed from the ms.

- Figure legends:

1. Please note that the exact p-values are not provided in the legends of figures 1C, E, F, G, I, J, K, L; 2C, D, F, H, I, L; 3A, E, H, K; 4C, F; 5C, D, E, H, I; 6A, EV2 C, EV3 B, D; EV4 A-E. Please provide exact p-values as reasonable.

Response:

Exact p-values of above Figure panels were provided in corresponding figure legends.

2. Please indicate the statistical test used for data analysis in the legends of figures EV1 B, EV4 A-E

Response:

Statistical test method used for Figures EV1 B, EV4 A-E have been described in legends.

3. Please note that information related to n is missing in the legends of figures EV3 B, D; EV4 A-E

Response:

Number of independent experiments was provided in the legends of figures EV3 B, D; EV4 A-E.

4. Please note that the error bars are not defined in the legends of figures EV4 A-E

Response:

The error bars of figures EV4 A-E were defined in corresponding legends.

EMBO press papers are accompanied online by A) a short (1-2 sentences) summary of the findings and their significance, B) 2-3 bullet points highlighting key results and C) a synopsis image that is exactly 550 pixels wide and 200-600 pixels high (the height is variable). The synopsis image should provide a sketch of the major findings, like a graphical abstract. Please note that text needs to be readable at the final size. Please send us this information along with the final manuscript.

Response:

Summary of finding:

STING inhibits LINE-1 retrotransposition to maintain the genomic stability via an IFN-independent mechanism. LINE-1 ORF1p interacts with STING and sorting ORF1p to lysosomes for degradation.

Bullet points:

- 1) STING inhibits LINE-1 retrotransposition independently of cGAS and IFN production.
- 2) STING inhibition of LINE-1 retrotransposition requires STING's trafficking out of ER.
- 3) STING-mediated LINE-1 ORF1p degradation requires sorting to endolysosomes.

We decided to use the original Fig 6F as the Synopsis image.

I would like to suggest some minor changes to the abstract. Do you agree with:

The cyclic dinucleotide sensor stimulator of interferon (IFN) genes (STING) is known for its critical role in interferon and inflammatory responses. In addition, STING also has functions independent of interferon induction. In this study, we report that STING restricts the mobilization of the cellular retrotransposon long interspersed nuclear element 1 (LINE-1) independent of cGAS and interferon induction. LINE-1 is the only active autonomous retrotransposable element in the human genome and its transposition can cause genetic and autoimmune diseases. STING inhibition of LINE-1 requires its dimerization. Mechanistically, STING interacts with LINE-1 ORF1p, then the complex translocates to the ER-Golgi intermediate compartment (ERGIC) and the Golgi followed by sorting to Rab7-positive lysosomes for degradation. Our data unveil a function of STING in maintaining host genome integrity by restricting LINE-1 retrotransposition via an IFN-independent mechanism.

Response:

We sincerely appreciate and agree with the editor's constructive revise on the manuscript's abstract section and have incorporated the revised abstract into ms file.

Referee #1:

I appreciate the authors' effort in revising the manuscript. However, the study now concludes without a clear explanation of how LINE-1 is degraded by STING. Additionally, there is a surprising requirement for the cGAMP-binding pocket, despite cGAMP having no impact. I recommend another revision to address the following two questions, which, if answered, would significantly improve the impact of the paper.

1. The claim that "STING trafficking is crucial for the suppression of LINE-1 by STING" is in question. The STING trafficking-deficient mutant LR333/334AA is as potent as STING-WT in ORF1p regulation (Fig 5F, 6B), suggesting that the regulation is independent of STING trafficking from the ER to the Golgi. ESCRT knockdown (siALIX+siTSG101, see PMID 29622626) will be a quick test to find out if microautophagy is essential.

Response:

We agree with the reviewer suggestion. As recommended by reviewers, we performed individual and combinatorial knockdown of the ESCRT nucleation factors ALIX and TSG101 to assess their effects on STING-mediated regulation of LINE-1 retrotransposition. The corresponding results were presented at Appendix Fig. S7 and described in lines 308-315, "Recent studies reported that STING is degraded by the endosomal sorting complexes required for transport (ESCRT)-driven lysosomal microautophagy to prevent hyperactivation(Balka *et al*, 2023a; Gentili *et al*, 2023; Kuchitsu *et al*, 2023). We silenced the ESCRT nucleating factors TSG101 and ALIX to explore the role of STING in LINE-1 regulation (Appendix Fig. S7A). STING retained its capacity to reduce ORF1p levels (Appendix Fig. S7B and S7C) and inhibit CMV-L1-neo^{RT} reporter retrotransposition (Appendix Fig. S7D and S7E) upon either single or combined knockdown of TSG101 and ALIX. These results indicate that LINE-1 ORF1p degradation by STING is independent of ESCRT-related lysosomal microautophagy."

Furthermore, we re-discussed the results that STING maintains its LINE-1 inhibitory activity even with the 333/334 mutations in lines 417-424, “BFA treatment, which causes the collapse of Golgi, partially restored ORF1p levels in the presence of STING (Fig. 4E and 4F), supporting a role of the canonical ER-to-Golgi trafficking pathway in ORF1p degradation by STING. In previous study, the STING mutant LR333/334AA loses interaction with SEC24C upon cGAMP stimulation, thus cannot use COPII vesicles to exit ER (Gui *et al.*, 2019). The interaction between ORF1 and STING might facilitate ER-to-Golgi transport through a non-canonical STING/SEC24C-independent pathway. Alternatively, the classical ER-Golgi pathway may not be the exclusive mechanism for ORF1p degradation in lysosomes, as ER-phagy could also mediate ORF1 degradation(Liao *et al.*, 2024; Sun *et al.*, 2023).”

Additionally, the STING-mediated degradation mechanism of ORF1p was re-discussed in lines 425-436, “The STING-ORF1p co-localized complex did not associate with LC3II, and STING-mediated inhibition of LINE-1 retrotransposition was unaffected upon depletion of key autophagy-related proteins (ULK1, ATG9a, Beclin1, p62, or ATG5). These results demonstrate that STING suppresses LINE-1 independently of both canonical and non-canonical macroautophagy. Furthermore, knockdown of core ESCRT components (TSG101 and ALIX) did not impair LINE-1 inhibition of STING, ruling out the involvement of ESCRT-dependent microautophagy. Surprisingly, individual knockout of TSG101 was found to impair the retrotransposition level of the LINE-1 reporter system, suggesting that TSG101 may play an auxiliary role in LINE-1 retrotransposition. The underlying mechanism remains unclear but appears to be independent of STING-mediated suppression of LINE-1 retrotransposition activity. Critically, the STING-ORF1p complex co-localized with the late endosome marker Rab7 and the lysosome marker LAMP1. These results collectively establish that STING targets ORF1p for degradation via the acidified endolysosomes.”

2. Line 376, "In addition, LINE-1 inhibition by STING is independent of cGAS but requires cGAMP binding site RY238/240". This is still rather surprising, unless the cGAMP-binding pocket also binds to LINE-1. Please assess if the cGAMP-binding mutant of STING RY238/240AA still associates with LINE-1 in co-IP. This would be a better control than the dimerization mutant (Appendix Fig. S4D) for the co-IP experiment as without dimerization, the protein is likely misfolded.

Response:

Thank you for the suggestion, we have included RY238/240AA mutant of STING to perform co-IP experiment, the result showed that the RY238/240AA mutant of STING was significantly impaired in its binding to ORF1p (Appendix Figure S4E). This result was described in lines 260-262, “This interaction was RNA- and cGAMP-independent (Appendix Fig. S4B and S4C), but required STING dimerization domain 153-173 and RY238/240 cGAMP-binding site (Appendix Fig. S4D and S4E).” and discussed in lines 401-405, “The co-IP results demonstrated that the RY238/240AA mutation attenuated the STING-ORF1p interaction (Appendix Fig. S4E), indicating that these residues may disrupt the binding interface required for STING-ORF1p complex formation while the precise underlying mechanism requires further investigation.”

Minor:

1. The only evidence to support a requirement of Golgi trafficking was BFA treatment which is quite toxic. It is thus over interpretation to claim that "STING trafficking is crucial for the suppression of LINE-1 by STING". Major point 1, if works, would answer the mechanism of

LINE1 degradation by STING.

Response:

Regarding the cytotoxicity of BFA, we determined the working concentration based on previous literature (Gonugunta *et al*, 2017; Gui *et al.*, 2019), and 2 μ M was found to be relatively moderate in HeLa cells. As suggested by the reviewer, we included knockdown experiments of the core ESCRT molecules ALIX and TSG101. The corresponding results are presented in Appendix Fig. S7 and described in the Major point 1 section. Additionally, we replaced the original Fig. 4E with a more representative image which quantified the colocalization coefficients of STING/ORF1p with the ER, ERGIC, and Golgi apparatus, and accordingly revised the conclusions of Fig. 4E in lines 247 to 250, “When we used brefeldin A (BFA) to disrupt the Golgi, STING and ORF1p were sequestered at ER and ERGIC but not colocalized with Golgi (Fig. 4E), and LINE-1 ORF1 protein expression was less inhibited by STING under BFA treatment (Fig. 4F). These results suggested that STING-mediated degradation of ORF1p requires a trafficking process from the ER to the Golgi.”

2. Figure 4E still has low image quality and lacks statistic support. As the requirement for STING trafficking in LINE1 degradation is weak, please either remove the whole panel or improve it (add colocalization quantification from 50-100 cells/condition). Similar for Fig 6D/E to ensure reproducibility.

Response:

Thank you for the suggestion. We have quantified 50 cells in every condition of fig 4E and Fig6D/E and replaced original images with ones with higher quality to improve data credibility and reproducibility rather than remove these data.

To determine three-color fluorescence colocalization coefficients, we first employed the Trainable Weka Segmentation plugin in ImageJ to identify ORF1p-STING colocalization regions, and subsequently calculated their colocalization coefficients with respective organelle markers (calnexin, ERGIC53, GM130, Rab7 and LAMP1). The analytical method for three-channel colocalization quantification was described in Figure legend of Fig. 4E in lines 938-941, and Fig. 6D and 6E in lines 1002-1005.

According to the statistical results, we redescribed Fig. 4E in lines 247-248, “When we used brefeldin A (BFA) to disrupt the Golgi, STING and ORF1p were sequestered at ER and ERGIC but not colocalized with Golgi (Fig. 4E)”. Fig. 6D and 6E were also quantified and described in lines 316-320, “Lastly, we performed immunofluorescence staining to detect colocalization of STING and ORF1p by lysosomal markers Rab7 and LAMP1. We observed colocalization of ORF1p/STING with the late endosome marker Rab7 and the lysosome marker LAMP1, which was enhanced by BafA1 treatment, suggesting that ORF1p/STING complex is sorted to lysosomes for degradation (Fig. 6D and 6E).”

3. lines 230 to 236, "To avoid cGAMP production from plasmid DNA transfection, we generated a HeLa cell line that stably expresses STING RY238/240AA...". Note that the cGAMP-binding deficient STING mutant cannot avoid cGAMP production by cGAS, but it can avoid the impact of cGAMP to STING. please reword accordingly.

Response:

Thank you for reviewer’s valuable corrections. We have corrected the content at lines of 230-231, “To avoid impact of cGAMP to STING activity, we generated a HeLa cell line that stably expresses STING RY238/240AA and assessed the expression of endogenous ORF1p.”

Referee #2:

The authors have addressed some of my concerns, but several critical issues remain unresolved. These concerns must be carefully addressed before the manuscript can be considered for publication.

1. There is an inconsistency in the γ H2AX levels presented in Figures 2J-K. Western blotting shows no significant change in γ H2AX following L1 overexpression, whereas immunofluorescence indicates a marked upregulation. How can this discrepancy be explained? Additionally, it would be beneficial to provide quantification for Figure 2J.

Response:

The inconsistency in the γ H2AX levels between the Western blot results (Fig. 2J) and IF (Fig 2K and 2L) may arise from the different experimental design. The WB assay displayed the overall γ H2AX in the entire cell population, which included both L1-transfected cells and un-transfected cells due to limited transfection efficiency. In contrast, for the IF experiments, in the Mock + L1 group, we specifically analyzed and quantified nuclear γ H2AX foci only in L1-transfected cells that exhibited ORF1p expression. Consequently, the increase in γ H2AX signal was more pronounced following L1-transfection in the IF experiments compared with WB.

We have also added grayscale quantitative analysis for Fig 2J.

2. The discussion regarding STING's role in DNA repair is not accurate. A previous study has demonstrated that STING promotes DNA repair, suggesting that the impact on γ H2AX levels in STING KO cells may not be solely attributed to its regulation on L1. This is further supported by the observation of increased γ H2AX levels in STING KO cells in the absence of L1 overexpression, as shown in Figure 2J.

Response:

We agree with the reviewer's suggestion and refined the discussion about the relationship between LINE-1 inhibition of STING and genomic damage repair. Relevant content was described in lines 346-353, "The sources of genomic damage are diverse, and LINE-1 retrotransposon activity is merely one contributing factor. In addition, STING exhibits repair activity against genotoxic stressors such as chemotherapy-induced DNA damage in cancer cells, and influences tumor relapse and progression. In cell-based experiments without an overexpressed LINE-1 reporter system, both STING knockout and overexpression have been shown to modulate the accumulation of γ H2AX foci number in the nucleus (Fig. 2K and 2L, Fig. EV2B and EV2C). This indicates that STING regulates genomic stability not only by inhibiting LINE-1 transposition, but also other mechanisms that affect genomic damage repair."

3. In the first round of review, both Reviewer 1 and I raised concerns about the claimed cGAS-independent but cGAMP-mediated function in ORF1p degradation. Given that cGAS is responsible for the synthesis of cGAMP and cGAS is dispensable for ORF1p degradation, why is cGAMP-mediated oligomerization of STING considered indispensable? The authors have yet to provide compelling experimental evidence or discussion on this contradictory point.

Response:

In previous study, we demonstrated that STING-mediated suppression of LINE-1 retrotransposition occurs independently of cGAS and cGAMP, but requires the cGAMP-binding

residues RY238/240 (Fig. 3A, 3B, 5C and Fig. EV3). Following the reviewers' suggestions, we performed co-immunoprecipitation (co-IP) assays to examine how the 238/240AA mutation affects STING-ORF1p interaction. As demonstrated by our experimental results, the 238/240AA residues are indeed critical for maintaining STING's binding activity with ORF1p (Appendix Figure S4E), this data was described in lines 260-262, "This interaction was RNA- and cGAMP-independent (Appendix Fig. S4B and S4C), but required STING dimerization domain 153-173 and RY238/240 cGAMP-binding site (Appendix Fig. S4D and S4E)", and discussed in lines of 396-405, "In addition, LINE-1 inhibition by STING is independent of cGAS but requires cGAMP binding site RY238/240. This is supported by the decreased expression of ORF1p in cells stably expressing the wild type STING but not the RY238/240 mutant (Fig. EV3A and EV3C). Similarly, LINE-1 activity is suppressed in cell lines expressing wild type STING wild-type but not its mutant RY238/240AA (Fig. EV3B and EV3D). These results rule out the effect of endogenous cGAMP and suggest that STING-mediated suppression of LINE-1 does not require the presence of cGAMP. The co-IP results demonstrated that the RY238/240AA mutation attenuated the STING-ORF1p interaction (Appendix Fig. S4E), indicating that these residues may disrupt the binding interface required for STING-ORF1p complex formation while the precise underlying mechanism requires further investigation."

Dear Prof. Guo,

Thank you for the submission of your revised manuscript. We have now received the enclosed reports from the referees. Both of them have some more comments that I would like you to incorporate before we can proceed with the official acceptance of your manuscript.

Please provide a point-by-point response with your final ms file that explains how you have addressed the last concerns.

When you upload a new version of your ms on our website you can bring forward all old files and then only replace the files that need to be replaced.

Referee #1:

I have no more concerns. The new co-IP data supports a role of STING's cGAMP-binding sites in ORF1p interaction, which resolves the contradictory interpretations in the last version of the paper. The Appendix Fig S4 provides key insights into the mechanism of STING-ORF1p interaction, which should be moved to a major figure. Please remove cGAMP from the Synopsis image, because this function is independent of cGAMP and cGAS.

Referee #2:

In the current manuscript, the authors claimed that "LINE-1 inhibition by STING requires the RY238/240 motif, not the presence of cGAMP". Since cGAMP binding triggers STING dimerization, if cGAMP is dispensable, dimerization should likewise be dispensable. However, the authors' data suggested that STING-mediated LINE-1 inhibition depends on the RY238/240 motif but not dimerization itself. I agree with the authors' hypothesis that the RY238/240AA mutation likely disrupts the binding interface necessary for STING-ORF1p complex formation. However, this contradicts their repeated assertions that dimerization is critical- e.g., "STING inhibition of LINE-1 requires its dimerization" and "ER localization, CDN binding, and dimerization are required for STING-mediated LINE-1 suppression". These conclusions are confusing.

Response to comments on EMBO Reports submission EMBOR-2024-60709V3. "STING inhibits LINE-1 retrotransposition through sorting ORF1p to lysosomes for degradation."

We thank the editor and reviewers for their valuable comments, and provide a point-by-point response and revisions included in this manuscript, as outlined below.

Referee #1:

I have no more concerns. The new co-IP data supports a role of STING's cGAMP-binding sites in ORF1p interaction, which resolves the contradictory interpretations in the last version of the paper. The Appendix Fig S4 provides key insights into the mechanism of STING-ORF1p interaction, which should be moved to a major figure. Please remove cGAMP from the Synopsis image, because this function is independent of cGAMP and

cGAS.

Response:

We appreciate the reviewer's thoughtful comment. In response, we have relocated Appendix Figures S4 to become main Figures 5A to 5E. Furthermore, we have removed the cGAMP from both the Synopsis image and Figure 6F to accord with the results we have presented.

Referee #2:

In the current manuscript, the authors claimed that "LINE-1 inhibition by STING requires the RY238/240 motif, not the presence of cGAMP". Since cGAMP binding triggers STING dimerization, if cGAMP is dispensable, dimerization should likewise be dispensable. However, the authors' data suggested that STING-mediated LINE-1 inhibition depends on the RY238/240 motif but not dimerization itself. I agree with the authors' hypothesis that the RY238/240AA mutation likely disrupts the binding interface necessary for

STING-ORF1p complex formation. However, this contradicts their repeated assertions that dimerization is critical-e.g., "STING inhibition of LINE-1 requires its dimerization" and "ER localization, CDN binding, and dimerization are required for STING-mediated LINE-1 suppression". These conclusions are confusing.

Response:

Thank you for the valuable comment. We have checked the manuscript and found that there was an ambiguous description of relationship with STING dimerization and CDN binding. Under steady state conditions, STING forms a dimer and locates at ER. The dimerization occurs independently of cGAMP activation. We have redescribed relevant content at introduction in lines 39-41, "Under steady state conditions, STING forms a dimer. Upon cGAMP binding, the STING dimer undergoes a conformational switch and promotes the lateral oligomerization, and then traffics to the Golgi apparatus(Burdette *et al*, 2011; Ishikawa & Barber, 2008).", and at results in lines 216-220, "ER bound STING forms a dimer under the steady-state condition (Ouyang *et al*, 2012; Shang *et al*, 2019). We therefore tested the effects of STING mutant Δ 153-173 (defective in dimerization)(Ouyang *et al.*, 2012), RIR/AIA (defective in ER location) (Sun *et al*, 2009) and RY238/240AA (defective in CDN binding) (Gui *et al.*, 2019; Zhang *et al*, 2019) on its ability to suppress LINE-1 retrotransposition activity."

We also modified conclusion of Fig4A to 4D in lines 227-229, "These data suggest that ER localization, dimerization and cGAMP binding site are required for STING-mediated LINE-1 suppression." to emphasize the sequential order of STING dimerization and cGAMP binding.

In subsequent sections, we further elaborate on the mechanistic rationale for STING's ER localization and dimerization in LINE-1 suppression, along with additional characterization of the RY238/240 residue sites in lines 230-235, "Since STING constitutively localizes to the endoplasmic reticulum (ER) as a dimer under physiological conditions, the requirement of its ER-targeting RIR motif and dimerization motif (residues 153-173) for LINE-1 suppression is mechanistically plausible. However, given that STING-mediated LINE-1 inhibition occurs independently of cGAS, the observed impact of the cGAMP-binding site on this activity presents an intriguing paradox. We therefore sought to investigate whether STING-dependent LINE-1 suppression requires the presence of cGAMP."

Prof. Fei Guo
Institute of Pathogen Biology, and Center for AIDS Research, Chinese Academy of Medical Sciences & Peking Union Medical College
Dongdan Santiao 9
Beijing 100730
China

Dear Prof. Guo,

I am very pleased to accept your manuscript for publication in the next available issue of EMBO reports. Thank you for your contribution to our journal.

Yours sincerely,
